EMBO
Molecular Medicine

# SHP-1 agonist SC-43 limits methicillin-resistant *Staphylococcus aureus* infection through inhibition of heme biosynthesis

Yini Huang[1,2,3,8], Yan Ye[4,8], Xinmei Zhu[1,5], Dengpan Liang[1,2,3], Ruiqin Cui[6], Xiaopeng Yuan[1,3], Xitao Li[7✉], Quanming Zou[4✉], Haibo Li [ID][4✉] & Wei Huang [ID][1,2,3✉]

## Abstract

**The limitations of existing drugs and the development of drug resistance make it urgent to develop new drugs against methicillin-resistant *Staphylococcus aureus* (MRSA). The re-development of the antibacterial activity of drugs that have already been proven safe for human use is an effective way. In this study, we discovered that the Src homology region 2 domain-containing phosphatase-1 (SHP-1) agonist SC-43, exhibits potent activity against Gram-positive bacteria, including MRSA. The mode of action studies revealed that SC-43 inhibits the key enzyme coproporphyrin ferrochelatase (CpfC) of the coproporphyrin-dependent (CPD) heme synthesis pathway and interferes with the bacterial porphyrin metabolism. The determination of the structure of CpfC derived from *S. aureus* (SA_CpfC) in this study allowed us to reveal the inhibitory effect of SC-43 at the molecular level. Animal experiments showed that SC-43 has the potential to become a new anti-MRSA drug. In conclusion, this study discovered a new anti-MRSA activity of a drug currently undergoing clinical trials and simultaneously verified the feasibility of developing new anti-Gram-positive bacteria drugs by inhibiting the CPD pathway.**

Subject Categories Microbiology, Virology & Host Pathogen Interaction; Pharmacology & Drug Discovery

## Introduction

Methicillin-resistant *Staphylococcus aureus* (MRSA) has been identified as one of the most prevalent drug-resistant pathogens in high-income countries and the leading causes of healthcare-associated and community-acquired infections worldwide (Antimicrobial Resistance Collaborators, 2022; European Antimicrobial Resistance Collaborators, 2022). In line with the high estimated burden, MRSA is still included in the high-priority pathogen category of world health organization (WHO) bacterial priority pathogens list (WHO, 2024).

Vancomycin and linezolid are recommended as first-line agents for the treatment of MRSA by the European and American guidelines (Nazli et al, 2024; Chastre et al, 2014; Mandell et al, 2007). However, the emergence of vancomycin-resistant *S. aureus* (VRSA) strains and thrombocytopenia caused by vancomycin and linezolid have limited its application (Nazli et al, 2024; Yang et al, 2023; Lee and Caffrey, 2017). In addition, linezolid can also cause severe side effects, including optic and peripheral neuropathy and lactic acidosis, which affects the compliance of the patient (Lee and Caffrey, 2017). Other anti-MRSA drugs, including daptomycin, also have limitations such as insufficient local tissue distribution (Nazli et al, 2024; Swift et al, 2021; Liu et al, 2024). As a result, for complicated MRSA infections, there is currently a very restricted range of therapeutic options. It is imperative and urgent to develop new anti-MRSA drugs that reduce resistance and side effects (WHO, 2024).

Drug repurposing is an effective way to discover new alternatives for different diseases, including MRSA infection. Recently, many cases (pinaverium bromide, benzbromarone, doxifluridine, visomitin, salifungin, 3-acetyl-11-keto-beta-boswellic acid and fenoprofen) have been reported as drug repurposing for MRSA treatment (Meng et al, 2024; Zhang et al, 2024; Wu et al, 2024; Wang et al, 2024; Li and Ma, 2024; Jiang et al, 2023). These

[1]Department of Laboratory Medicine, Shenzhen Key Laboratory of Pathogenic Microbiology and Antimicrobial Resistance Surveillance, Shenzhen People's Hospital, (The First Affiliated Hospital, Southern University of Science and Technology; The Second Clinical Medical College, Jinan University), Shenzhen 518020, China. [2]Department of Pulmonary and Critical Care Medicine, Shenzhen Key Laboratory of Respiratory Diseases, Shenzhen Clinical Research Center for Respiratory Disease, Shenzhen Institute of Respiratory Diseases, Shenzhen People's Hospital (The First Affiliated Hospital, Southern University of Science and Technology; The Second Clinical Medical College, Jinan University), Shenzhen 518020, China. [3]Guangdong Provincial Clinical Research Center for Laboratory Medicine, Guangzhou 510220, China. [4]National Engineering Research Center of Immunological Products, Department of Microbiology and Biochemical Pharmacy, College of Pharmacy, Army Medical University, Chongqing 400038, China. [5]College of Life Science, Bengbu Medical University, Bengbu 233030, China. [6]Department of Pharmacy, Shenzhen People's Hospital (The First Affiliated Hospital, School of Medicine, Southern University of Science and Technology), Shenzhen 518020, China. [7]School of Pharmaceutical Sciences (Shenzhen), Shenzhen Campus of Sun Yat-sen University, Shenzhen 518107, China. [8]These authors contributed equally: Yini Huang, Yan Ye. ✉E-mail: lixt78@mail.sysu.edu.cn; qmzou@tmmu.edu.cn; lihaibo@tmmu.edu.cn; weihuang@mail.sustech.edu.cn

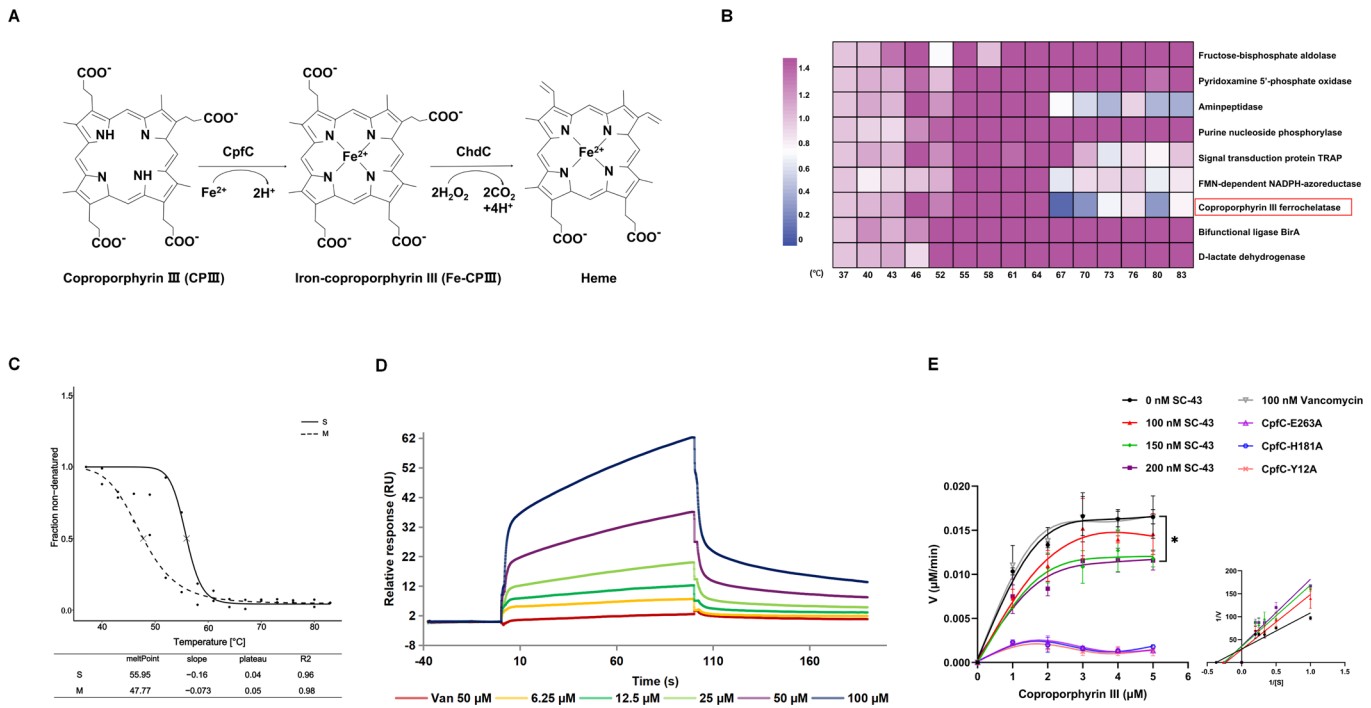

**Figure 1. The inhibitory effect of SC-43 on the key enzyme CpfC in the coproporphyrin-dependent (CPD) pathway.**

(A) Steps 2 and 3 of the CPD heme biosynthesis pathway. (B) The fold changes of proteins with significant changes in thermal stability after treatment with SC-43 (0.5 µg/mL) under different temperatures. (C) Melting curves for CpfC comparing DMSO (use M to represent) and SC-43 (0.5 µg/mL) (use S to represent) treatment. (D) Surface plasmon resonance (SPR) binding kinetic traces of the interactions of SC-43/vancomycin and CpfC. (E) Kinetic analysis of the inhibitory effect of SC-43 on CpfC ferrochelatase activity and the ferrochelatase activity of CpfC with Y12A, H181A, or E263A point mutations. 4 µM CpfC was assayed in the presence of varying concentrations of coproporphyrin III (CPIII) and a fixed concentration of 0.7 nM of iron. Michaelis–Menten and Lineweaver–Burk plots (insets), curves were generated using the kinetic constants obtained from nonlinear regression. 0 vs 200 nM, $^*P = 0.0412$. Error bars indicate mean ± standard deviation from biological replicates ($n = 3$). Two-way ANOVA, $^*P < 0.05$. Source data are available online for this figure.

candidates are expected to become new treatments for MRSA. However, given the high failure rate in the development of new drugs, more relevant attempts and studies are still needed.

SC-43 is a previously identified Src homology region 2 domain-containing phosphatase-1 (SHP-1) agonist with anti-tumor and anti-fibrotic effects (Hu et al, 2017; Su et al, 2017). In this study, we described the discovery of a novel anti-MRSA activity of SC-43. The mechanism of action studies showed that SC-43 exerts its antibacterial effect by inhibiting the function of coproporphyrin ferrochelatase (CpfC), thereby interfering with the biosynthesis of heme (Fig. 1A). In addition, the structure of CpfC derived from *S. aureus* (SA_{CpfC}) was determined in this study, which allowed us to reveal the inhibitory effect of SC-43 at the molecular level.

## Results

### Antibiogram of SC-43

Based on the antibacterial phenotype, we screened the compound library that was in clinical trials at phases 1–3. We found that SC-43 exhibited the lowest antibacterial activity against *S. aureus* American Type Culture Collection (ATCC) 29213 (SA_{29213}) (minimum inhibitory concentration (MIC) <0.625 µg/mL) (Appendix Table S1). Antibacterial susceptibility tests for more species

showed that SC-43 exhibits potent activity against Gram-positive bacteria, including MRSA and its clinical isolates, while no activity against Gram-negative bacteria (Table 1). Among them, the activity against strains of SA_{29213}, *Enterococcus faecalis* ATCC 29212, MRSA 252, and clinical isolates of MRSA, methicillin-susceptible *S. aureus* (MSSA), vancomycin-resistant *Enterococcus faecium* (VRE), penicillin-susceptible *Streptococcus pneumoniae* (PSSP) and methicillin-resistant *Staphylococcus epidermidis* (MRSE) are superior to vancomycin, which is currently the most used clinically against these pathogens. In addition, we combined SC-43 with Polymyxin B nonapeptide (PBNA), which can destroy the outer membrane but has no antibacterial activity, and found that it was still ineffective against Gram-negative bacteria, indicating that the ineffectiveness of SC-43 is not due to the barrier of the outer membrane (Table 1). SC-43 may have a unique target for Gram-positive bacteria only. However, the MIC values of SC-43 for *A. baumannii* and *P. aeruginosa* decreased to 12.5 and 3.13 µg/mL, respectively. This indicates that there may be a synergistic effect between SC-43 and the outer membrane disruptors.

### Profile of proteins with potential interaction with SC-43 in *S. aureus*

SC-43 is an orally active Sorafenib derivative with potent anti-fibrotic and anti-cancer effects, which are exerted by SHP-1

**Table 1. The antibacterial activity of SC-43.**

| Strains/Isolates[a] | SC-43 | MIC (µg/mL) SC-43 (10 µg/mL PBNA[b]) | Vancomycin |
|---|---|---|---|
| *S. aureus* ATCC 29213 | 0.2 | --[c] | 1 |
| SA$_{29213: \, cpfC}$[d] | 3.2 | -- | 1 |
| SA$_{29213: \, pyJ335}$[d] | 0.2 | -- | 1 |
| *A. baumannii* ATCC 19606 | >50 | 12.5 | -- |
| *K. pneumoniae* ATCC 13883 | >50 | >50 | -- |
| *P. aeruginosa* ATCC 27853 | >50 | 3.13 | -- |
| *E. coli* ATCC 25922 | >50 | >50 | -- |
| *E. faecalis* ATCC 29212 | 1.6 | -- | 4 |
| MRSA 252 | 0.4 | -- | 1 |
| CC49050 | 0.2 | -- | 1 |
| CC48973 | 0.2 | -- | 1 |
| CC42266 | 0.2 | -- | 32 |
| CCF3993 | 1.6 | -- | 4 |
| CC26110 | 0.2 | -- | 2 |

[a]All American Type Culture Collection (ATCC) reference strains and clinical isolates were obtained from the collection of the clinical microbiology laboratory of Shenzhen People's Hospital. The clinical isolates CC48973, CC49050, CC42266, CCF3993, and CC26110 were methicillin-resistant *S. aureus* (MRSA), methicillin-susceptible *S. aureus* (MSSA), vancomycin-resistant *Enterococcus faecium* (VRE), penicillin-susceptible *S. pneumoniae* (PSSP) and methicillin-resistant *Staphylococcus epidermidis* (MRSE), respectively.
[b]Polymyxin B nonapeptide.
[c]Not applicable.
[d]The *S. aureus* ATCC 29213 strain with *cpfC* overexpression or vector.

dependent inhibition of signal transducer and activator of transcription 3 (STAT3) (Hu et al, 2017; Su et al, 2017). It is obvious that the above mechanism of action is absent in bacteria, suggesting that SC-43 has another target and mechanism of action in its antibacterial activity. After comparing the activities of SC-43 against various pathogens, *S. aureus*, which showed the lowest minimum inhibitory concentration (MIC), was selected for the target identification study. With the aim of discovering the target of SC-43, thermal proteome profiling (TPP) was used to assess the thermal stability and interactions by measuring thermal denaturation across the proteome. A total of 1886 proteins were identified, of which nine had a significantly increased thermal stability after the treatment (Dataset EV1; Fig. 1B), indicating that SC-43 may interact with them.

Based on the antibacterial activity and antibacterial spectrum of SC-43, we mainly rely on two aspects to determine whether a protein is a potential target. Firstly, it must be an essential protein for Gram-positive bacteria used in this study. Secondly, it is very likely to be non-essential for Gram-negative bacteria (the possibility that SC-43 is ineffective against Gram-negative bacteria due to the obstruction of the outer membrane was ruled out by using SC-43 with an outer membrane disruptor). Among the proteins whose thermal stability was affected, CpfC was considered a potential antibacterial target of SC-43. CpfC catalyzes the penultimate step of coproporphyrin-dependent (CPD) heme synthesis, where $Fe^{2+}$ is inserted into coproporphyrin III (CPIII) (Fig. 1A) (Layer, 2021).

Since monoderm bacteria, including *S. aureus*, can only utilize the CPD pathway to synthesize heme (Dailey and Medlock, 2022), and SC-43 appears to be effective against Gram-positive bacteria only, it is reasonable to further investigate CpfC as a target of SC-43. From the melting curve of TPP experiment, we observed that the melting temperature ($T_m$) value of CpfC increased by 8.18 °C after SC-43 treatment (Fig. 1C). However, starting from 67 °C, after the treatment, the fold changes of CpfC protein did not seem to show any regular pattern (Fig. 1B). This is mainly because at 67 °C or higher temperature, even under the treatment of SC-43, the vast majority (>90%) of the CpfC protein has undergone denaturation. The extremely low amount of remaining soluble CpfC protein may be the reason for the unstable trend of its changes (Fig. 1C). We then constructed an overexpression strain of *cpfC* and conducted AST on it. The results showed that the overexpression of *cpfC* led to a 16-fold increase in the MIC of SC-43, suggesting that its antibacterial activity is related to *cpfC* (Table 1).

## Interaction between SC-43 and SA$_{CpfC}$

To further clarify the interaction between SC-43 and SA$_{CpfC}$, we first produced SA$_{CpfC}$ recombinant protein (Appendix Fig. S1) and evaluated whether there was direct binding between them by surface plasmon resonance (SPR). The results showed that SC-43 was directly bound to SA$_{CpfC}$ with high affinity $K_D = (7.17 \pm 1.28) \times 10^{-5}$ M, while the control drug vancomycin did not show any binding (Fig. 1D). In addition, the ferrochelatase activity assay showed that SC-43 inhibited SA$_{CpfC}$ in a dose-dependent manner, indicating that the direct binding ultimately affects the function of SA$_{CpfC}$ (Fig. 1E).

## SC-43 inhibits bound heme biosynthesis and ATP levels in *S. aureus*

After identifying the inhibitory effect on SA$_{CpfC}$ function, it was logical to infer that SC-43 would affect the biosynthesis of iron-coproporphyrin III (Fe-CPIII) and heme (bound heme) biosynthesis. We utilized the property that bound heme are prone to lose their metal components under the condition of hot acidulous environment to evaluate the free porphyrins and bound heme in bacteria after SC-43 treatment. The results showed that after SC-43 treatment, the levels of bound heme significantly decreased (Fig. 2A). Given that CpfC is responsible for catalyzing the chelating of $Fe^{2+}$ to the porphyrin ring, the above results are consistent with the phenotype of CpfC being inhibited.

As a cofactor of various enzymes, heme plays an irreplaceable role in physiological processes such as respiration and oxidative stress responses (Poulos, 2014). In this study, we observed that the intracellular ATP content was significantly decreased after SC-43 treatment (Fig. 2B).

## Exogenous heme supplementation compensates for the inhibitory effect of SC-43, which can cause bacterial hyperpolarization

To further clarify the relationship between heme biosynthesis and the antibacterial activity of SC-43, we compared the growth of *S. aureus* in the medium with or without heme. We found that after adding exogenous heme to the medium, the inhibitory effect of SC-43 against *S.*

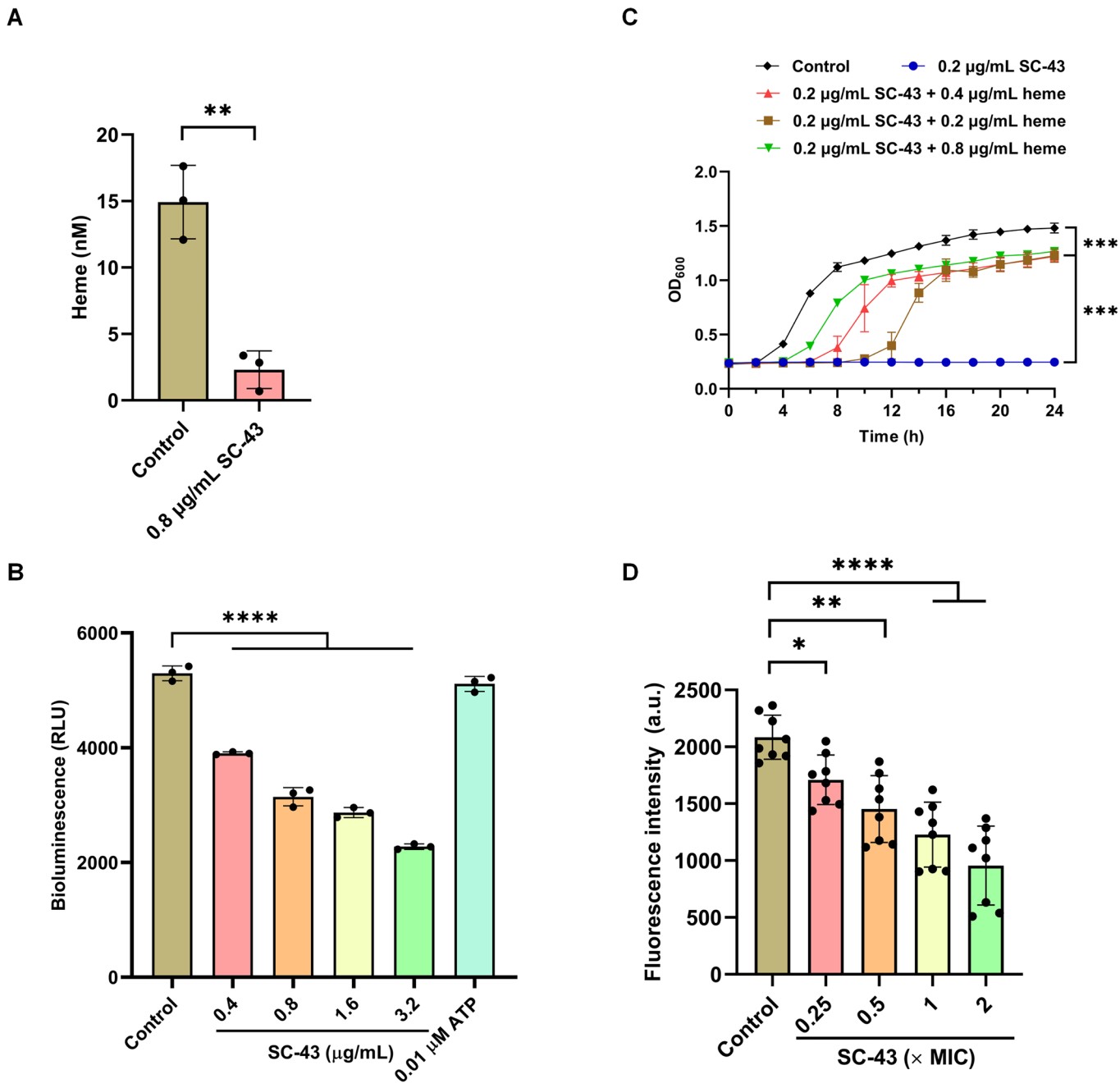

**Figure 2. The effects of SC-43 on heme biosynthesis and related functions.**

(A) $Fe^{2+}$-bound porphyrins in *S. aureus* ATCC 29213 ($SA_{29213}$) after treatment with SC-43 (0.8 μg/mL), paired *t*-test. Control vs SC-43, $^{**}P = 0.0022$. (B) The intracellular ATP levels of $SA_{29213}$ after treatment with varying concentrations of SC-43. Control vs 0.4 μg/mL, $^{****}P < 0.0001$; 0.8 μg/mL, $^{****}P < 0.0001$; 1.6 μg/mL, $^{****}P < 0.0001$; 3.2 μg/mL, $^{****}P < 0.0001$. (C) The growth curves of $SA_{29213}$ in the medium in the presence or absence of varying concentrations of heme. Optical density at 600 nm was monitored at the indicated time points. Control vs 0.2 μg/mL SC-43, $^{***}P = 0.0001$; 0.2 μg/mL SC-43 vs 0.2 μg/mL SC-43 + 0.2 μg/mL heme, $^{***}P = 0.0009$; (D) The change in the membrane potential of $SA_{29213}$ incubated with varying concentrations of SC-43, indicated by fluorescence of the dye DiSC3(5) (0.5 μM). a.u. arbitrary units. Control vs 0.25 × MIC, $^{*}P = 0.0112$; 0.5 × MIC, $^{**}P = 0.0012$; 1 × MIC, $^{****}P < 0.0001$; 2 × MIC, $^{****}P < 0.0001$. Error bars indicate mean ± standard deviation from biological replicates ($n = 3$ for (A–C), $n = 8$ for (D)). One-way ANOVA, $^{****}P < 0.0001$, $^{***}P < 0.001$, $^{**}P < 0.01$ and $^{*}P < 0.05$. Source data are available online for this figure.

*aureus* was significantly compensated (Fig. 2C). This result strongly confirms that the inhibition of endogenous heme synthesis is the mechanism by which SC-43 exerts its activity. In addition, we also observed that SC-43 can cause hyperpolarization in *S. aureus* (Fig. 2D). These are consistent with the phenotype of heme biosynthesis inhibition.

## Identification of differential metabolites

Untargeted metabolomics was used to investigate differences in metabolomic signatures after SC-43 treatment. A total of 1230 and 775 metabolites were obtained in both electrospray ionization (ESI)

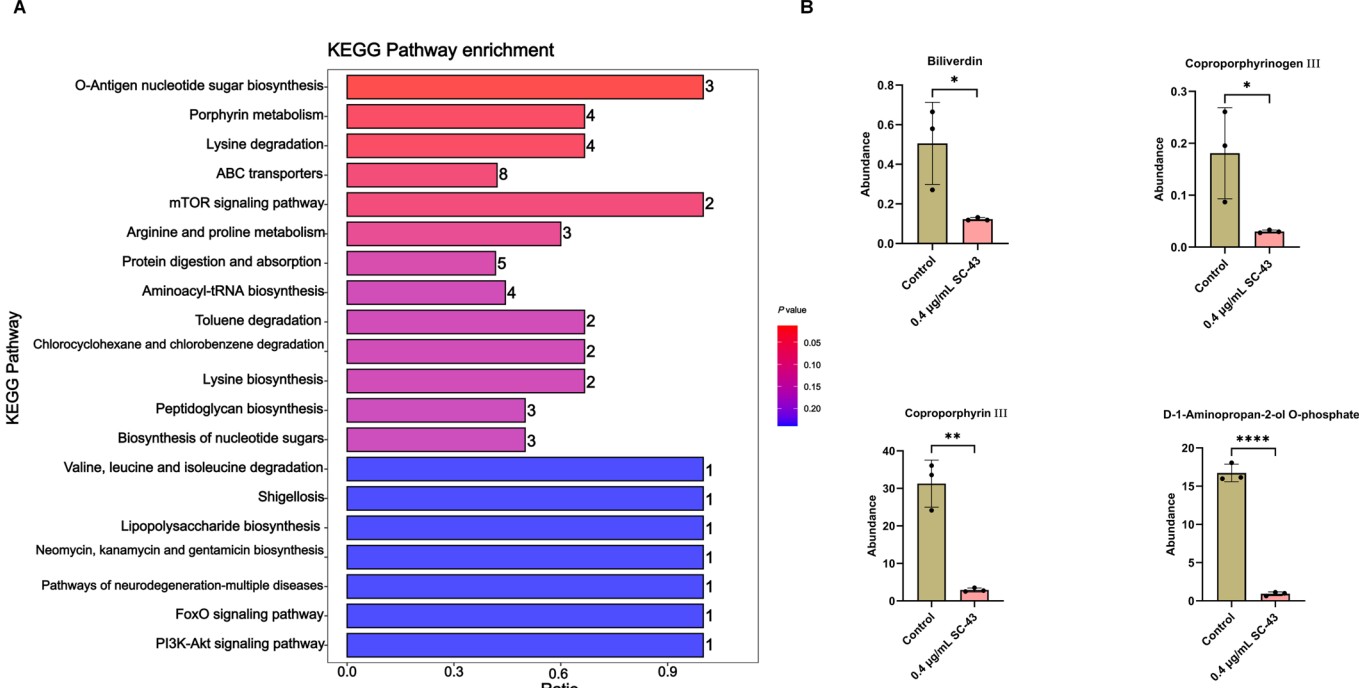

**Figure 3. The effects of SC-43 on the metabolism of *S. aureus*.**

(A) Kyoto Encyclopedia of Genes and Genomes (KEGG) enrichment analysis illustrating the top pathways that have been enriched by differentially expressed metabolites after SC-43 treatment. The number of differentially expressed metabolites enriched in the corresponding pathways are displayed on the right side of the box. (B) The comparison of four differentially altered metabolites in the porphyrin metabolism pathway. Control vs 0.2 μg/mL SC-43, biliverdin, $^*P = 0.0331$; coproporphyrinogen III, $^*P = 0.0408$; coproporphyrin III, $^{**}P = 0.0015$; D-1-aminopropan-2-ol O-phosphate, $^{****}P < 0.0001$. Error bars indicate mean ± standard deviation from biological replicates ($n = 3$). Hypergeometric test for (A), unpaired *t*-test for (B), $^{****}P < 0.0001$, $^{**}P < 0.01$, and $^*P < 0.05$. Source data are available online for this figure.

positive and ESI negative, respectively (Dataset EV2). From these metabolites, 512 differential metabolites were identified, among which 220 were upregulated, and 292 were downregulated after SC-43 treatment. When categorized by Kyoto Encyclopedia of Genes and Genomes (KEGG) pathways, O-antigen nucleotide sugar biosynthesis, lysine degradation and porphyrin metabolism were significantly enriched (Dataset EV2; Fig. 3A). In the porphyrin metabolic pathway, the production of biliverdin, coproporphyrinogen III, CPIII and D-1-aminopropan-2-ol O-phosphate were significantly reduced after the SC-43 treatment (Fig. 3B). The metabolites that changed in the other pathways are listed in Dataset EV2.

## SC-43 is predicted to impair the enzymatic function by competitively occupying its active site

To clarify the molecular mechanism of the interaction between SC-43 and $SA_{CpfC}$, we first obtained $SA_{CpfC}$ crystals that diffracted to a resolution of 2.19 Å (PDB ID: 9VI5). The crystal structure revealed that $SA_{CpfC}$ belonged to the P1 space group (complete data collection and refinement statistics are provided in Appendix Table S2). The sequence alignment result showed that $SA_{CpfC}$ and CpfC of *Listeria monocytogenes* ($LM_{CpfC}$) share 60% similarity (Fig. 4A). A high structural conservation was also observed at the tertiary level (Fig. 4B). The monomeric structure of $SA_{CpfC}$ contains two ferredoxin-like domains, each containing a four-stranded parallel

β-sheet flanked by α-helices. The enzymatically active pocket of $SA_{CpfC}$ harbors several conserved residues: Histidine 181 (H181) (corresponding to H182 of $LM_{CpfC}$), glutamic acid 263 (E263) (corresponding to E263 of $LM_{CpfC}$), and the proximal tyrosine 12 (Y12) (corresponding to Y12 of $LM_{CpfC}$). Previous reports confirmed that these residues are mainly responsible for the interaction between CpfC and the porphyrin ring, which is critical for catalysis (Andrea et al, 2023).

## Intermolecular interactions within the SC-43-$SA_{CpfC}$ complex

Molecular docking was conducted using the Glide module of Schrödinger to elucidate the binding mode of SC-43 with $SA_{CpfC}$. The crystal structure of $SA_{CpfC}$ resolved in this study was processed with the Protein Preparation Wizard in Schrödinger, and SC-43 was docked into the heme-binding site using Glide's extra precision (XP) mode. The highest-ranked binding pose was selected for molecular dynamics (MD) simulation to evaluate the stability and intermolecular interactions within the SC-43-$SA_{CpfC}$ complex. A 200 ns MD simulation was performed using the Desmond module of Schrödinger, with the system achieving equilibrium after approximately 60 ns, as evidenced by the stabilization of the root-mean-square deviation (RMSD) of the protein-ligand complex (Fig. 5A). Subsequent analysis therefore focused on the 60–200 ns trajectory segment.

**A**

**B**

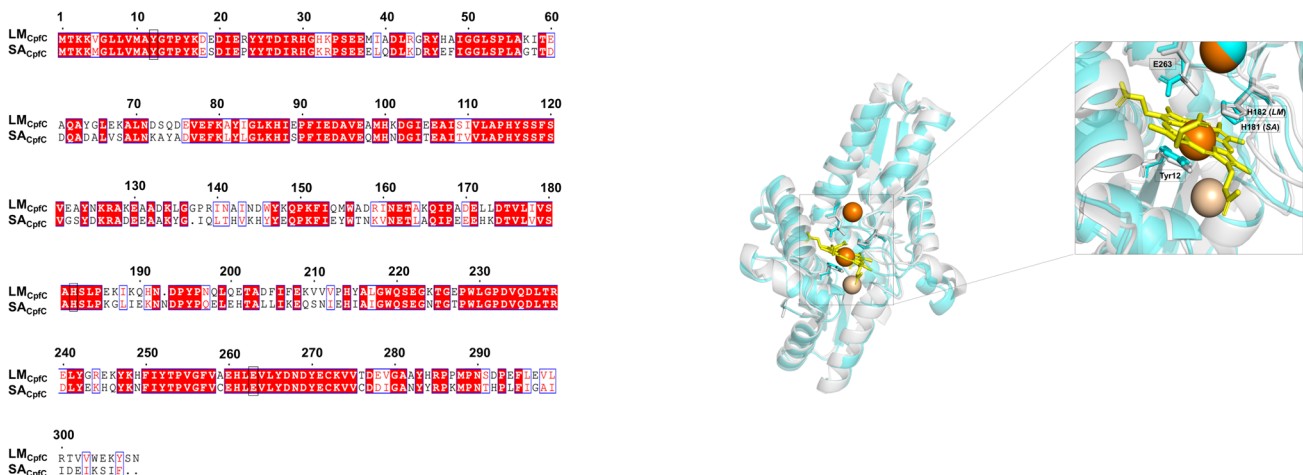

**Figure 4. The comparison of CpfC derived from *S. aureus* (SA_CpfC) and *L. monocytogenes* (LM_CpfC).**

(A) The sequence alignment of SA_CpfC and LM_CpfC. (B) The comparative analysis of structure between SA_CpfC (PDB ID: 9VI5) and LM_CpfC (PDB ID: 9F0F) (Cyan represents SA_CpfC, gray represents LM_CpfC, yellow represents coproporphyrin III (CPIII), brown represents $Fe^{2+}$, and pale yellow represents $Cl^-$). Source data are available online for this figure.

During this timeframe, as illustrated in Fig. 5B and further detailed in two-dimensional and three-dimensional (Fig. 5C,D) schematic diagrams, Y12 established a hydrogen bond with the urea carbonyl group of SC-43 for 88% of the analyzed period and a π-π stacking interaction with the terminal phenyl ring of SC-43 for 80% of the time. Additionally, the two urea NH groups of SC-43 consistently formed hydrogen bonds with E263 throughout most of the trajectory. Furthermore, π-π stacking interactions were observed between the central phenyl ring of SC-43 and Y25 for 36% of the time, and between the terminal phenyl ring and H181 for 47% of the time. These interactions are hypothesized to stabilize the SC-43-SA_CpfC complex. Furthermore, we were unable to detect the catalytic function of iron chelation for the Y12A, H181A and E263A mutant proteins, thereby confirming the significance of these amino acid sites (Fig. 1E).

To validate these findings, we produced Y12A, H181A and E263A mutant proteins of SA_CpfC, respectively. Compared with the wildtype of protein, both SPR and molecular docking analyses demonstrated that the binding affinity of the Y12A, H181A and E263A mutant proteins to SC-43 was significantly reduced (Dataset EV3; Table 2, Appendix Fig. S2). These results further corroborate our hypothesis that SC-43 impairs SA_CpfC function through competitively binding to the active site.

### Efficacy of SC-43 in a murine model of infection

The in vivo activity of SC-43 was evaluated using a mouse model of MRSA skin wound infection (Fig. 6A). Compared with the same dose of mupirocin, SC-43 treatment significantly reduced the wound size (Fig. 6B,C) and local bacterial load (Fig. 6D). On day 7 post-treatment, bacteria were almost undetectable in the locally infected tissues of SC-43-treated mice.

## Discussion

As an effective way to develop new antibacterial agents, drug repurposing has been widely used in the development of anti-MRSA drugs (Abavisani et al, 2025). We here report a novel activity of the SHP-1 agonist SC-43 against Gram-positive bacteria in vitro and in vivo. Compared to vancomycin, the current first-line drug for MRSA, SC-43 has lower MIC values. SC-43 also showed good activity against VRE, indicating that it can be used as an effective supplement when vancomycin is inefficient in the clinic.

Drugs previously used for other diseases often have different mechanisms of action from existing antibiotics due to their structural differences, which is more advantageous for them to overcome existing resistance mechanisms (Abavisani et al, 2025). This is critical for MRSA, which is associated with high mortality and morbidity and has developed multiple drug resistance (Nazli et al, 2024). The brand-new mechanism of SC-43, which inhibits CpfC and thereby interferes with heme synthesis, indicates that it can bypass the existing resistance mechanisms and meets the requirements for new drug development for MRSA.

Heme is an essential enzyme cofactor that plays important roles in many metabolic pathways, including respiration, gas sensing, and detoxification of reactive oxygen species (Dailey and Medlock, 2022). The strategy of inhibiting the growth of organisms by interfering with the synthesis of heme has previously been demonstrated in plants. Herbicides that inhibit plant growth by inhibiting protoporphyrinogen oxidase have been widely used (Hao et al, 2011). In eukaryotes, the use of the heme synthesis enzyme ferrochelatase inhibitor in the treatment of ocular neovascularization has also been reported recently (Sishtla et al, 2022). In 2015, it was first discovered that eukaryotic cells and Gram-negative bacteria share the conserved protoporphyrin-dependent (PPD)

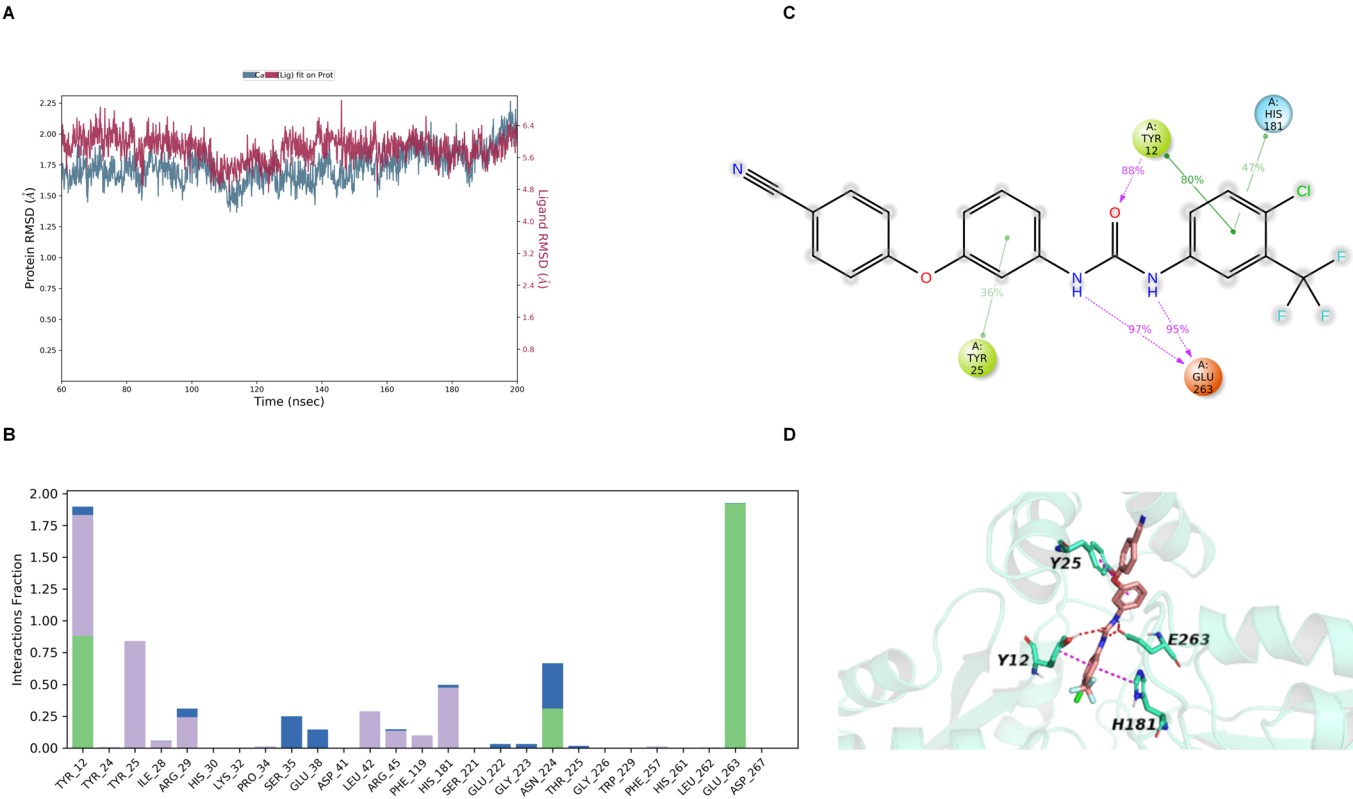

**Figure 5. Molecular dynamics simulation of the interaction between SC-43 and SA$_{CpfC}$.**

(A) RMSD plots for the protein SA$_{CpfC}$ and the ligand SC-43 over the simulation period; (B) Time-course analysis of protein-ligand contacts; (C) Detailed schematic representation of atomic interactions between the ligand and specific protein residues; (D) Three-dimensional visualization of the ligand (carbon atoms in wheat) interacting with the protein (carbon atoms in greencyan); hydrogen bonds are shown as red dashed lines, and π–π interactions are indicated by magenta dashed lines. Source data are available online for this figure.

**Table 2. Surface plasmon resonance (SPR) analysis of the interaction between SC-43 and SA$_{CpfC}$ wild-type and mutants.**

| $K_D$ (M) | | | |
|---|---|---|---|
| SA$_{CpfC}$ | SA$_{CpfC\ (Y12A)}$[a] | SA$_{CpfC\ (H181A)}$[a] | SA$_{CpfC\ (E263A)}$[a] |
| $(7.17 \pm 1.28) \times 10^{-5}$ | $(3.09 \pm 1.34) \times 10^{-4}$ | $(2.27 \pm 1.77) \times 10^{-4}$ | $(2.15 \pm 0.59) \times 10^{-4}$ |

[a]Mean ± standard deviation from three biological replicates. One-way ANOVA.
*$P < 0.05$.

pathway for the last three steps of heme synthesis, while Gram-positive bacteria employ the CPD pathway (Dailey et al, 2015; Lobo et al, 2015). In addition, there is also the sirohaem pathway that is widely distributed among bacteria (Videira et al, 2020). Therefore, it may not be an ideal strategy to develop antibacterial agents against Gram-negative bacteria from the perspective of inhibiting heme synthesis due to the possibility of side effects in humans. On the contrary, it is a promising strategy for Gram-positive bacteria. Gram-positive bacteria have a unique set of terminal reactions, which are CgoX-catalyzed oxidation of coproporphyrinogen III, CpfC-catalyzed Fe$^{2+}$ insertion of CPIII, and ChdC-catalyzed formation of heme (Dailey et al, 2015). Therefore, CgoX, CpfC and ChdC are ideal targets for the development of drugs that are specifically against Gram-positive bacteria and are likely to have a good safety profile in humans. Meanwhile, some of the key

influencing factors of the aforementioned pathway, such as IsdG that can interact with CpfC and affect the biosynthesis of heme, are also considered as ideal targets for the treatment of *S. aureus* infections (Videira et al, 2018; Almeida et al, 2025).

Recently, it has been identified that butylcycloheptylprodiginine (BCHP) and prodigiosin can specifically target the CPD pathway by screening the compound library using the *E. coli* YFP strain carrying the CPD pathway genes of *S. aureus*. Molecular docking results showed the binding interactions between BCHP and *L. monocytogenes* coproheme decarboxylase (Lm-ChdC) (Jackson et al, 2023). In this study, we discovered an inhibitor, SC-43, for another key enzyme CpfC in the CPD pathway, and verified that it exhibits antibacterial activity by inhibiting heme synthesis (Fig. 7). However, since heme also serves as an essential iron source, we cannot rule out the possibility that the inhibition of bacterial

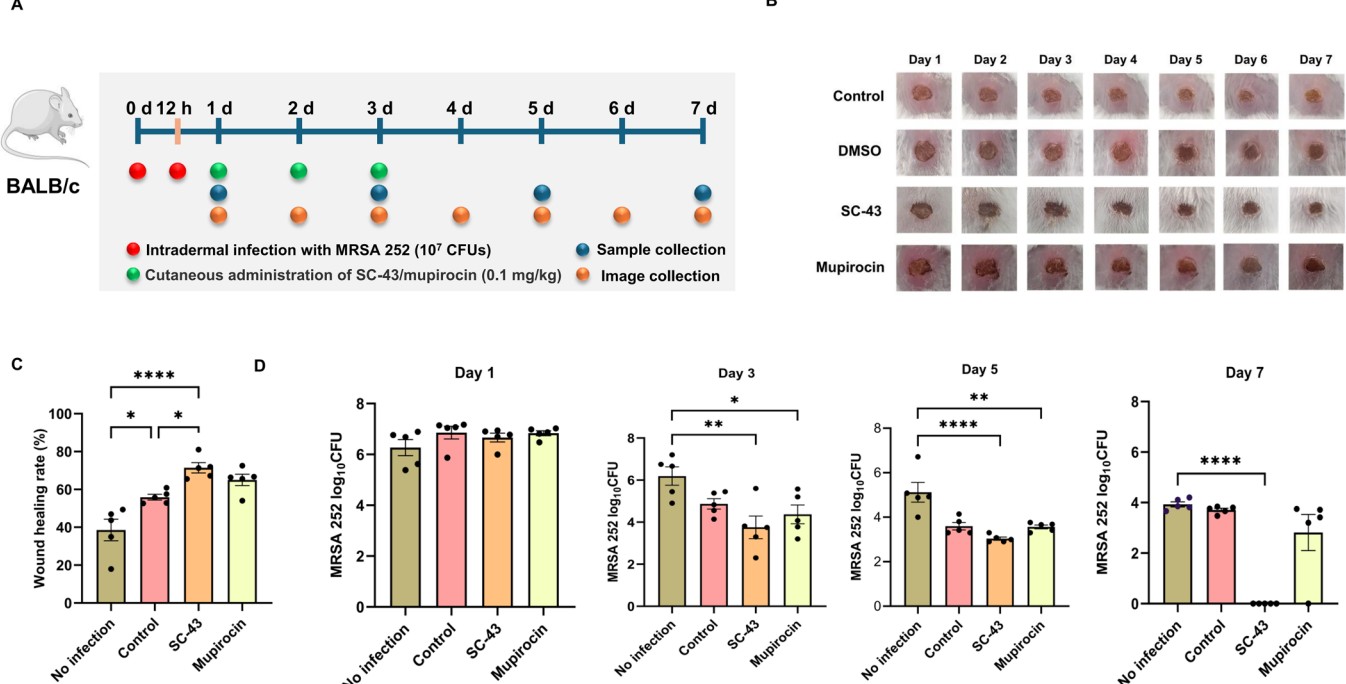

**Figure 6. SC-43 exhibits potent antibacterial activity in a mouse model of MRSA wound infection.**

(A) The dosing schedule and route of drug administration for the mouse model. (B) The size of infected wounds in mice treated with DMSO, SC-43 (0.1 mg/kg), and mupirocin (0.1 mg/kg) ($n = 20$ per group). (C) The comparison of the size of the original wound with that of the wound on day 7. No infection vs Control, $^*P = 0.0190$; No infection vs SC-43, $^{****}P < 0.0001$; Control vs SC-43, $^*P = 0.0111$; (D) Bacterial loading of the wound site in mice treated with DMSO, SC-43 (0.1 mg/kg), and mupirocin (0.1 mg/kg) were recorded continuously. At day 3, No infection vs SC-43, $^{**}P = 0.0077$, No infection vs mupirocin, $^*P = 0.0189$; At day 5, No infection vs SC-43, $^{****}P < 0.0001$; No infection vs mupirocin, $^{**}P = 0.0086$; At day 7, No infection vs SC-43, $^{****}P < 0.0001$; Error bars indicate mean ± standard deviation ($n = 5$ per group); One-way ANOVA, $^{****}P < 0.0001$, $^{**}P < 0.01$, and $^*P < 0.05$. Source data are available online for this figure.

growth is caused by the deficiency of iron due to the reduction of heme (Hoffmann et al, 2025). The other potential roles of heme may also contribute to the inhibition of bacterial survival (Dailey and Medlock, 2022). Moreover, metabolomics results indicate that SC-43 reduces the content of coproporphyrin precursors rather than causing accumulation (Fig. 3B). We speculate that this might be a compensatory measure taken by the bacteria due to the adverse effects of the accumulation of porphyrin metabolites on their survival.

The failure to obtain the co-crystallization structure of SA$_{CpfC}$-SC-43 is a limitation of our work. Regarding the potential binding sites in CpfC that were identified through molecular docking, and which might interact with SC-43, we were unable to obtain the strains with the corresponding point mutations after numerous attempts by using gene editing methods, which prevented us from verifying the MICs of SC-43 in these strains. In addition, through a standard passaging experiment, no spontaneous mutant strain resistant to SC-43 was obtained. This indicates that bacteria are not prone to developing resistance to it, and at the same time, it also hinders our acquisition of genetic evidence. Given that the mutations at these sites have a significant impact on the enzymatic activity of CpfC (Fig. 1E), this might explain why spontaneous drug-resistant strains and genetic evidence could not be obtained.

It is worth noting that, apart from CpfC, among the proteins whose thermal stability was affected, there are some that are also crucial for bacterial survival under specific conditions (Fig. 1B). For example, fructose-bisphosphate aldolase is a conserved and crucial glycolytic enzyme and is also regarded as a potential pharmacological target (Capodagli et al, 2014). However, there are two classes of fructose-bisphosphate aldolase present in *S. aureus*, and the protein identified by the TPP experiment in this study belongs to class I (Dataset EV1). Therefore, the presence of class II aldolase makes it highly unlikely that the identified fructose-bisphosphate aldolase in this study is the target responsible for the antibacterial effect. The effect of SC-43 on fructose-bisphosphate aldolase class I may be beneficial for the clearance of the pathogen under harsh external conditions, such as the immune pressure that bacteria encounter when infecting the host. For the other proteins identified by the TPP experiment, it is still worth further investigation to determine whether they have any other pharmacological significance under specific conditions. The results of metabolomics also indicate that, in addition to the porphyrin metabolic pathway closely related to CpfC, SC-43 can also affect O-antigen nucleotide sugar biosynthesis and lysine degradation pathways, suggesting its multiple effects on bacteria (Fig. 3A). The MICs drop in the presence of PBNA for *P. aeruginosa* and *A. baumannii* may be related to these metabolic pathways.

More importantly, the potent antibacterial activity demonstrated against clinical isolates of MRSA and VRE, as well as the results in animal models, have enabled us to validate the clinical

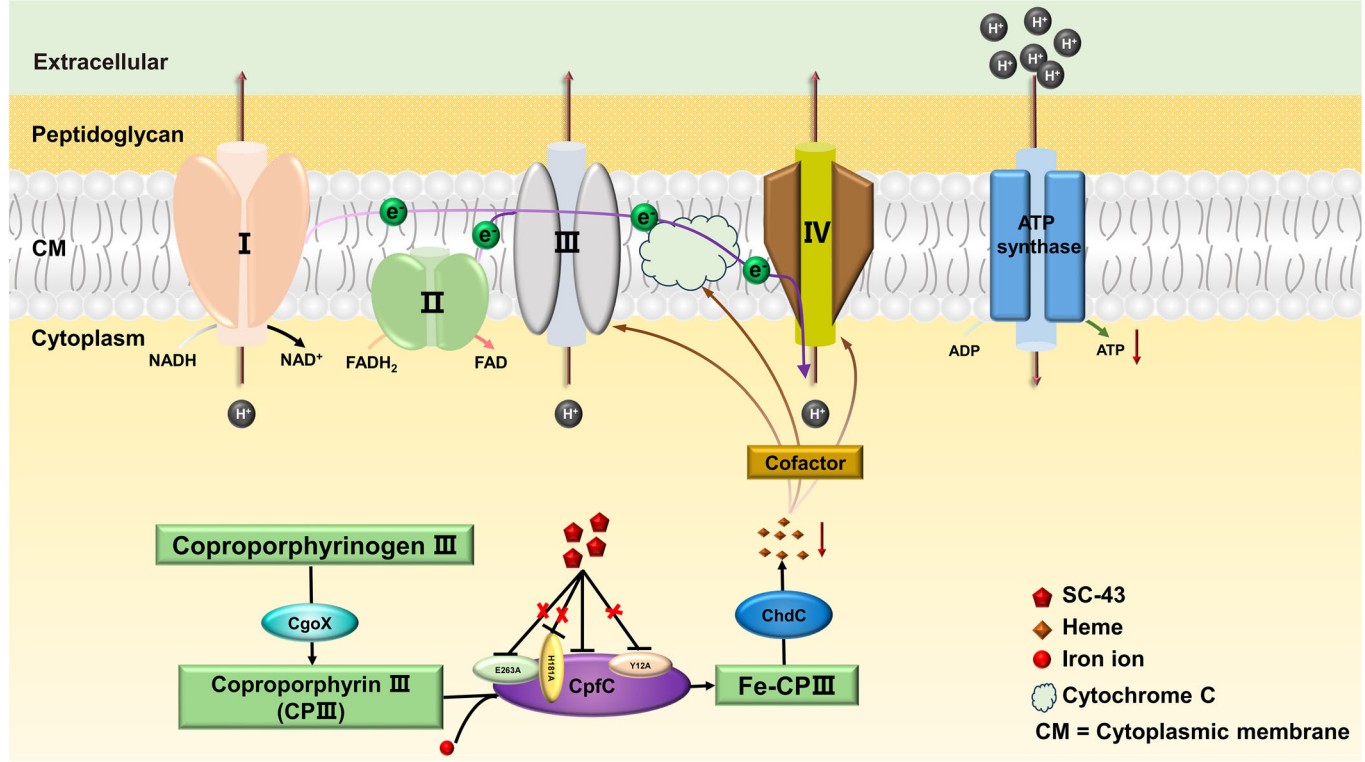

**Figure 7. Schematic diagram of the mechanism of SC-43 against MRSA.**

CpfC is a key enzyme in the process by which *S. aureus* synthesizes heme using the coproporphyrin-dependent (CPD) pathway, and it is responsible for catalyzing the insertion of iron ions into coproporphyrin III (CPIII). SC-43 can inhibit the activity of CpfC and affect the heme synthesis of *S. aureus*. Since heme is a cofactor of various complexes in the bacterial electron transport chain, SC-43 may potentially inhibit ATP synthesis by affecting the electron transport chain. The Y12, H181, and E263 of CpfC are the key sites that determine its interaction with SC-43.

application prospects of inhibiting the CPD pathway. SC-43 has been approved by the FDA for phase I clinical trial for refractory solid tumors (ClinicalTrials.gov ID: NCT03443622) and phase I/II clinical trial in combination with Cisplatin for advanced or refractory non-small cell lung cancer or biliary tract carcinoma (ClinicalTrials.gov ID: NCT04733521), which indicates that it already has very solid preclinical data to ensure its safety. The sequence alignment result showed that the identity between $SA_{CpfC}$ and human SHP-1 is 16.34%, and no conserved amino acids for catalyzing iron chelation were found in the SHP-1 sequence either (Appendix Fig. S3). This indicates that the CpfC of *S. aureus* and the SHP-1 of mammalian cells may be two independent target sites for SC-43. This study provides a new direction for the subsequent development of new antibiotics targeting Gram-positive bacteria and offers confidence for attracting more subsequent investment. However, we should also note that SC-43 can inhibit the growth of *S. epidermidis*, and the resulting impact on the normal skin flora cannot be ignored. Since exogenous heme can affect the antibacterial activity of SC-43, unlike topical applications, it is still unknown whether the heme present in serum and tissues will have an impact on the efficacy of the systemic applications.

In conclusion, this study extends our understanding of the pharmacological activities of SC-43. Although more experiments are still needed to verify the selectivity arguments of CPD vs PPD, we revalidate the feasibility and prospect of targeting key enzymes in the CPD pathway for the development of new drugs against Gram-positive bacteria, including MRSA.

## Methods

**Reagents and tools table**

| Reagent/resource | Reference or source | Identifier or catalog number |
| --- | --- | --- |
| **Experimental models** | | |
| BALB/c mice | Beijing HFK Bioscience | N/A |
| MRSA 252 | Zeng et al, (2023) | N/A |
| **Recombinant protein** | | |
| $SA_{CpfC}$ | Novopro | G0263371-1 |
| $SA_{CpfC\ (Y12A)}$ | Novopro | G0367851-1 |
| $SA_{CpfC\ (H181A)}$ | Novopro | G0353544-1 |
| $SA_{CpfC\ (E263A)}$ | Novopro | G0367851-2 |

| Reagent/resource | Reference or source | Identifier or catalog number |
|---|---|---|
| **Oligonucleotides and other sequence-based reagents** | | |
| PCR primers | This study | Methods |
| **Chemicals, enzymes, and other reagents** | | |
| Trypticase soy broth | BBI Life Science | B682007 |
| Luria Broth | Sangon Biotech | A507002 |
| Mueller–Hinton broth | OXOID | CM0405B |
| Horse blood | MeilunBio | MB2968 |
| Methyl thiazolyl tetrazolium | Diamond | A100793 |
| SC-43 | TargetMol | T8478 |
| Phosphate-buffered saline | BBI Life Science | E607008 |
| Dithiothreitol | Thermo Fisher | A300862-0005 |
| Iodoacetamide | Invitrogen | I10189 |
| Trypsin | Thermo Fisher | 90057 |
| Tandem mass tag | Thermo Fisher | 90064B |
| Acetonitrile | Thermo Fisher | CAS 75-05-8 |
| Formic acid | Pierce | 28905 |
| NdeI | NEB | R0111S |
| XhoI | NEB | R0146S |
| *E. coli* DH5α | Sangon Biotech | B528413 |
| *E. coli* BL21 | Sangon Biotech | B528419 |
| Kanamycin | TargetMol | T64922 |
| IPTG | BBI Life Science | CAS 367-93-1 |
| HEPES | Thermo Fisher | 1563630 |
| NaCl | Sangon Biotech | CAS 7647-14-5 |
| Imidazole | Sangon Biotech | CAS 288-32-4 |
| Glycerin | BBI Life Science | CAS 56-8-5 |
| EDTA | BBI Life Science | A350353-0100 |
| P20 | BBI Life science | CAS 9005-64-5 |
| Glycine | BBI Life Science | CAS 56-40-6 |
| CPIII | MCE | CAS 14643-66-4 |
| Triton X-100 | BBI Life Science | CAS 9002-93-1 |
| Acetone | Thermo Fisher | A949-4 |
| Porcine hemin | Sigma | A11165 |
| Tris-EDTA (pH 8.0) | Sangon Biotech | B548127-0500 |
| Tris-HCl (pH 7.5) | Sangon Biotech | B648002-0500 |
| ATP | Thermo Fisher | B300581-0250 |
| Mupirocin | TargetMol | T1465 |
| Sodium cholate | MeilunBio | MB1115 |
| Tween-80 | Solarbio | CAS 9005-65-6 |
| Mercaptoethanol | Macklin | CAS 60-24-2 |
| PEG300 | MCE | CAS 25322-68-3 |
| **Software** | | |
| PyMOL v2.5 | Open source | http://www.pymol.org/ |

| Reagent/resource | Reference or source | Identifier or catalog number |
|---|---|---|
| Coot v0.9.8.7 | MRC Laboratory | https://www2.mrc-lmb.cam.ac.uk |
| PHENIX v1.18.2 | PHENIX | https://phenix.jp |
| Desmond 2022 academic version | Schrodinger Life Science | https://www.schrodinger.com/ |
| Schrödinger's Simulation Interaction Diagram tool | Schrodinger Life Science | https://www.schrodinger.com/ |
| Desmond's trajectory clustering tool | Schrodinger Life Science | https://www.schrodinger.com/ |
| Schrödinger's Ligand Interaction Diagram tool | Schrodinger Life Science | https://www.schrodinger.com/ |
| Biacore 8 K Insight Evaluation v6.0 | GE Healthcare | N/A |
| Prism v8.0 | GraphPad | https://www.graphpad.com |
| Compound discoverer v3.3 | Thermo Fisher | N/A |
| MetaboAnalystR v4.0.0 | R | N/A |
| **Other** | | |
| CC48973 | Cheng et al, 2024 | NCBI: PRJNA1054306 |
| CC49050 | Cheng et al, 2024 | NCBI: PRJNA1054306 |
| CC42266 | Cheng et al, 2024 | NCBI: PRJNA1054306 |
| CCF3993 | Cheng et al, 2024 | NCBI: PRJNA1054306 |
| CC26110 | This study | N/A |
| SPR System | GE Healthcare | Biacore 8 K |
| LC-MS/MS | Thermo Fisher | ORBITRAP ECLIPSE |
| Microplate reader | BioTek | BioTek Synergy H1 |
| Amino coupling kit | Cytiva | BR100050 |
| BCA protein assay Kit | Thermo Fisher | 23227 |
| ATP Bioluminescence Assay Kit CLS II | Roche | 11699695001 |

## Bacterial strains

All ATCC reference strains and clinical isolates were obtained from the collection of the clinical microbiology laboratory of Shenzhen People's Hospital (Cheng et al, 2024). The clinical isolates CC48973, CC49050, CC42266, CCF3993, and CC26110 were MRSA, methicillin-susceptible *S. aureus* (MSSA), vancomycin-resistant *Enterococcus faecium* (VRE), penicillin-susceptible *S. pneumoniae* (PSSP), and methicillin-resistant *Staphylococcus epidermidis* (MRSE), respectively (Cheng et al, 2024). Apart from CC26110, the genome sequences of the clinical isolates in this study were deposited in the National Center for Biotechnology Information under the number PRJNA1054306.

## Medium and growth conditions

Trypticase soy broth (TSB) and Luria Broth (LB) and were used for the growth of Gram-positive and negative bacteria, respectively. Cation-adjusted Mueller–Hinton broth (CAMHB) was used for antibacterial susceptibility testing (AST). Mueller–Hinton agar (MHA) was used for bacterial colony-forming unit (CFU) enumeration. The addition of 3% lysed horse blood was performed for *S. pneumoniae*. Unless otherwise stated, bacteria were grown at 37 °C under aerobic conditions with shaking at $200 \times g$.

## AST assays

MICs were determined by broth microdilution in accordance with CLSI guidelines (Clinical and Laboratory Standards Institute [CLSI], 2024). The 50 µL bacterial suspension, equivalent to a 0.5 McFarland standard, was added to an equal volume of twofold serial dilution of the compound. Polypropylene 96-well microtiter plates (Corning, USA) with a final volume of 100 µL were incubated at 37 °C for 16–20 h. The Methyl thiazolyl tetrazolium (MTT) assay was used to evaluate bacterial growth. The MIC was defined as the lowest concentration of compounds that inhibited 90% of bacterial growth. All MIC determinations were performed in biological triplicate.

## Screening strategy

ASTs were performed using a phase 1–3 clinical compound library (Topscience, China) to identify the candidates capable of inhibiting $SA_{29213}$. The candidate with the most potent activity was selected for mechanism studies (Appendix Table S1 and Table 1).

## TPP assay

We employed the TPP approach to identify the target of SC-43 against *S. aureus*. SC-43 (0.5 µg/mL) or DMSO was added to the $SA_{29213}$ bacterial suspensions at the exponential phase ($OD_{600} = 0.5$), and then incubated at 37 °C for 30 min. The precipitates were harvested by centrifugation at $3000 \times g$ for 5 min and underwent double washes with precooled phosphate-buffered saline (PBS) containing either SC-43 at a concentration of 0.5 µg/mL or an equal volume of DMSO. After adjusting the bacterial suspensions to $OD_{600} = 10$, a 100 µL aliquot was transferred to a PCR tube, followed by centrifugation at $3000 \times g$ for 5 min. After discarding the supernatants, the precipitates were incubated at sequential temperatures (37, 40, 43, 46, 52, 55, 58, 61, 64, 67, 70, 73, 76, 80, and 83 °C) for 3 min each. The total protein was extracted with the Gram-positive bacterial protein extraction kit. Cells were lysed by sonication on ice (three cycles of 30 s pulse, 1 min rest) and centrifuged at $15,000 \times g$ for 15 min at 4 °C to remove cell debris. The supernatant (soluble proteome) was collected and quantified using a BCA protein assay. Soluble proteins were reduced with 10 mM dithiothreitol (DTT) at 56 °C for 30 min, alkylated with 55 mM iodoacetamide (IAA) in the dark for 30 min, and digested with sequencing-grade trypsin (1:50 w/w) overnight at 37 °C. The resulting peptides were labeled with tandem mass tag (TMT) reagents according to the manufacturer's protocol. Samples from different temperature points were pooled for multiplexed analysis. Labeled peptides were fractionated using high-pH reverse-phase chromatography and analyzed by liquid chromatogram-mass

spectrum/mass spectrum (LC-MS/MS) on an Orbitrap Fusion Lumos mass spectrometer coupled to an EASY-nLC 1200 system (Thermo Fisher, USA). Peptides were separated on a C18 column (75 µm × 25 cm) with a gradient of 5–35% acetonitrile in 0.1% formic acid over 120 min. MS data were acquired in data-dependent acquisition (DDA) mode with a top-speed method (3 s cycle time). Raw MS data were processed using MaxQuant with the *S. aureus* reference proteome database. TMT reporter ion intensities were extracted, and protein abundance was quantified. Thermal melting curves were generated for each protein by plotting normalized protein abundance against temperature. Tm was determined by fitting the data to a sigmoidal curve using a nonlinear regression model. Proteins with significantly increased in thermal stability upon SC-43 treatment were analyzed and visualized by using the R package TPP (v3.28.0) with the following rules: (i) The melting point difference between vehicle- and compound-treated conditions for a protein had a BH-corrected $P$ value of less than 0.05 and $\Delta Tm > 1$ °C (ii) The steepest slope of the protein melting curve in the vehicle- and compound-treated conditions pair was below $-0.06$ (Savitski et al, 2014; Franken et al, 2015).

## Gene overexpression

The gene encoding CpfC with flanked restriction endonuclease recognition sites for EcoRV was synthesized using the PCR-based accurate synthesis. EcoRV was used to digest DNA and *pyJ335*. The resulting products were ligated to form a recombinant plasmid *pyJ335-cpfC*. The recombinant plasmid was then transformed into *E. Coli* TOP 10 cells. After cloning, the plasmids were extracted and then further transformed via electroporation into *S. aureus* RN4220 and subsequently into $SA_{29213}$. The successful transformation of the *pyJ335-cpfC* plasmid was verified using PCR. The induction of gene expression was achieved by adding anhydrotetracycline (100 ng/mL). All the primer sequences were listed in Appendix Table S3.

## Production and purification of recombinant CpfC protein

The gene encoding $SA_{29213}$ CpfC was codon-optimized for expression in *E. coli* and synthesized commercially. The gene was amplified by PCR using primers designed to incorporate a 6 × His-tag at the N-terminus and restriction sites for cloning (NdeI and XhoI). The PCR product was ligated into the pET-30a(+) expression vector, which was then transformed into *E. coli* DH5α competent cells for plasmid propagation. Positive clones were verified by colony PCR and Sanger sequencing. The verified plasmid was transformed into *E. coli* BL21 (DE3) competent cells for protein expression. A single colony was inoculated into 10 mL of LB medium supplemented with 50 µg/mL kanamycin and grown overnight. The overnight culture was diluted 1:100 into 1 L of fresh LB medium with kanamycin and grown until $OD_{600} = 0.6$–0.8. Protein expression was induced by adding 0.5 mM isopropyl β-D-1-thiogalactopyranoside (IPTG), and the culture was incubated at 18 °C for 16–18 h with shaking at $180 \times g$ to promote soluble protein production. Cells were harvested by centrifugation at $4000 \times g$ for 20 min at 4 °C. The cell pellet was resuspended in lysis buffer (50 mM HEPEs, pH 7.5; 150 mM NaCl; 10 mM imidazole; 5% glycerin) and incubated on ice for 30 min. Cells were lysed by sonication on ice, and the lysate was clarified by centrifugation at

15,000 × g for 30 min at 4 °C. The supernatant was loaded onto pre-equilibrated Ni affinity column and the protein eluted using an imidazole gradient (10–500 mm). The eluted protein was then purified by size-exclusion chromatography using a Superdex 200 Increase column (GE Healthcare) equilibrated with buffer (25 mM HEPES, pH 7.5; 150 mM NaCl; and 2 mM DTT). Proteins were finally collected and concentrated to $A_{280} = 10$ and stored at −80 °C.

## Interaction between CpfC and SC-43

To verify that CpfC is the target of SC-43, we first used SPR experiments to clarify whether there is a direct binding between them. The recombinant CpfC protein was immobilized on a CM5 sensor chip using the amine coupling method. The CM5 chip surface was first activated with a 1:1 mixture of 0.4 M 1-ethyl-3-(3-dimethylaminopropyl) carbodiimide (EDC) and 0.1 M N-hydroxy succinimide (NHS) for 7 min at a flow rate of 10 μL/min. Following activation, the CpfC protein, diluted to 50 μg/mL in 10 mM sodium acetate buffer (pH 5.0), was injected over the chip surface for 7 min. Unreacted groups on the chip surface were deactivated by injecting 1 M ethanolamine-HCl (pH 8.5) for 7 min. The immobilization level was monitored in real-time, and a final immobilization level of ~10,000 response units (RU) was achieved. The running buffer used for all SPR experiments was HBS-EP+ (10 mM HEPES, 150 mM NaCl, 3 mM EDTA, and 0.05% v/v surfactant P20, pH 7.4). This buffer was filtered through a 0.22 μm membrane and degassed prior to use to prevent air bubble formation during the experiment. SC-43 was dissolved in DMSO to prepare a 10 mM stock solution. Working concentrations of SC-43 were prepared by diluting the stock solution in running buffer to final concentrations ranging from 6.25 to 100 μM. The final DMSO concentration in all samples was kept below 1% to avoid nonspecific binding effects. The interaction between CpfC and SC-43 was analyzed using a Biacore 8 K instrument (Cytiva, USA). The experiment was performed at 25 °C with a flow rate of 30 μL/min. SC-43 solutions were injected over the CpfC-immobilized surface for 120 s, followed by a dissociation phase of 180 s. The chip surface was regenerated between cycles using 10 mM glycine-HCl (pH 2.0) for 30 s to remove any residual bound analyte. A flow cell on the CM5 chip was activated and deactivated without protein immobilization to monitor nonspecific binding of SC-43 to the chip surface. Running buffer was injected over the CpfC-immobilized surface to account for any baseline drift or buffer effects. The sensorgrams were analyzed using Biacore 8 K Evaluation Software. The response from the blank surface and buffer controls was subtracted from the experimental data to account for nonspecific binding and baseline drift. The equilibrium dissociation constant ($K_D$) was determined by fitting the data to a 1:1 binding model.

## The effect of SC-43 on the activity of S. aureus CpfC

The activity of CpfC, $CpfC_{Y12A}$, $CpfC_{H181A}$, and $CpfC_{E263A}$ were evaluated by measuring the incorporation of $Fe^{2+}$ into CPIII to form Fe-CPIII, as previously described (Hobbs et al, 2017). Reactions were performed on a 96-well plate in a final volume of 100 μL. Each reaction contained 4 μM CpfC. To determine the kinetic parameters of CpfC in the presence and absence of SC-43, the enzyme activity was measured at varying concentrations of CPIII

(0–20 μM). Control reactions included 100 nM vancomycin (negative control), DMSO (vehicle control) and no enzyme (background control). Reactions were incubated at 37 °C for 30 min in the dark to prevent photodegradation of porphyrins. The formation of Fe-CPIII was monitored by measuring the absorbance at 390 nm using a microplate reader (BioTek, USA). The absorbance of Fe-CPIII was corrected by subtracting the background absorbance from control reactions. The initial reaction rates were plotted against substrate concentrations, and the data were fitted to the Michaelis–Menten equation. Reactions without SC-43 (vehicle control) to determine baseline activity and rule out solvent effects. Reactions without CpfC to account for non-enzymatic $Fe^{2+}$ incorporation.

## The effect of SC-43 on the biosynthesis of S. aureus- bound heme

$SA_{29213}$ bacterial suspensions at the exponential phase ($OD_{600} = 0.5$) were divided into control and SC-43-treated groups. SC-43 was added to the experimental groups at final concentrations of 0.8 μg/mL, while the control group received an equivalent volume of DMSO. Cultures were incubated for an additional 6 h before harvesting for heme extraction and quantification. Bacterial cells were harvested by centrifugation at 5000 × g for 10 min at 4 °C and washed twice with ice-cold PBS. The cell pellet was resuspended in 1 mL of lysis buffer (50 mM Tris-HCl, pH 7.5, 150 mM NaCl, and 1% Triton X-100) and subjected to sonication on ice (200 W, 2 s pulses, 3 s intervals, 5 min total duration) to disrupt the cells. An equal volume of acid-acetone solution (acetone:1 M HCl = 4:1, v/v) was added to the lysate, followed by vigorous vortexing and incubation at 4 °C in the dark for 30 min. Half of the mixture was taken and heated up to 95–98 °C for 30 min (for bound heme and free porphyrins level testing), during which the other half of the mixture was kept at room temperature (for free porphyrins level testing). The fluorescence of 200 μL clear supernatant with excitation at 400 nm and emission at 600 nm was monitored (Liu et al, 2014). A standard curve was generated using hemin chloride dissolved in DMSO at concentrations ranging from 1–100 nM.

## The effect of exogenous heme on the activity of SC-43 against MRSA

To clarify whether the inhibition of heme biosynthesis is related to the activity of SC-43 against MRSA. We compared the growth of the MRSA strain in the medium with or without porcine hemin (Sigma) (Hammer et al, 2013). MRSA 252 bacterial suspensions at exponential phase ($OD_{600} = 0.5$) were diluted 1:100 into fresh TSB with or without porcine hemin (0–0.8 μg/mL), and growth was measured by optical density at 600 nm every 2 h.

## The effect of SC-43 on the intracellular ATP levels

$SA_{29213}$ bacterial cultures at exponential phase ($OD_{600} = 0.5$) were continued to grow in the medium in the presence of SC-43 at 0.4, 0.8, 1.6, and 3.2 μg/mL for 1 h. After adjusting the bacterial concentration to $5 \times 10^6$ CFU/mL, a total volume of 1 mL culture was centrifuged at 5000 × g for 10 min to obtain the precipitate and resuspended with 50 μL of dilution buffer. The 450 uL of boiling 100 mM Tris, 4 mM EDTA, pH 7.75 were added to the cell

suspension, and incubated for another 2 min at 100 °C. The samples were centrifuged at $1000 \times g$ for 60 s. A volume of 50 uL supernatant was transferred to a black microplate to assay with 50 μL luciferase reagent. The intracellular ATP levels were measured using an ATP Bioluminescence Assay Kit CLS II (Roche, Switzerland) according to the manufacturer's instructions.

## Membrane polarization assay

SA_{29213} bacterial cells ($1 \times 10^8$ CFU/mL) in the exponential growth phase (OD_{600} = 0.6) were washed twice and resuspended in PBS. The cells were then stained with 0.5 μM DiSC3(5) (Aladdin) at 37 °C for 10 min. A volume of 190 μL stained bacteria and 10 μL of SC-43 or DMSO were transferred to a black 96-well plate (Corning, USA). After incubation at 37 °C for 30 min, fluorescence intensity was measured using an excitation wavelength at 622 nm and an emission wavelength at 670 nm using a microplate reader (BioTek, USA).

## Untargeted metabolomics

The overnight-cultured SA_{29213} was inoculated at a ratio of 1:100 into fresh TSB medium and cultured until an OD_{600} = 0.4. The bacterial cells were washed twice and resuspended by using an equal volume of fresh TSB in the presence or absence of SC-43 (0.4 μg/mL). After incubation at 37 °C for 30 min, the cells were centrifuged at $3000 \times g$ for 10 min to obtain the bacterial pellets. Use 400 μL lysis buffer (methanol: acetonitrile = 3:1) to lyse 30 mg of bacterial precipitate. After ultrasonic disruption on the ice for 10 min, the samples were left to stand at 4 °C for 2 h. They were then centrifuged at $10,000 \times g$ for 15 min at 4 °C. The supernatants were transferred into a new centrifuge tube and concentrated and dried using a vacuum freeze dryer. A volume of 100 μL 50% methanol water was used to re-dissolve the sample. The solutions were vortexed for 3 min and then centrifuged at $10,000 \times g$ for 15 min at 4 °C. The supernatants were transferred into the detection bottle for LC-MS detection. An equal amount of supernatant was taken from all samples and mixed into the QC samples for on-machine testing.

The LC analysis was performed on a UHPLC System. Chromatography was carried out with an ACQUITY UPLC® HSS T3 ($2.1 \times 100$ mm, 1.8 μm) (Waters, USA). The column was maintained at 40 °C. The flow rate and injection volume were set at 0.3 mL/min. For LC-ESI (+)-MS analysis, the mobile phases consisted of (A) 0.1% formic acid in water and (B) 0.1% formic acid in methanol. Separation was conducted under the following gradient: 0–0.5 min, 2% B ; 0.5–2 min, 2–50% B; 2–5 min, 50–98% B; 5–8 min, 98%B; 8–10 min, 98–2 B; 10–12 min, 2% B. For LC-ESI (-)-MS analysis, the mobile phases consisted of (A) 0.05% acetic acid in water and (B) 0.05% acetic acid in methanol. Separation was conducted under the following gradient: 0–0.5 min, 2% B; 0.5–2 min, 2–50% B; 2–5 min, 50–98% B; 5–8 min, 98%B; 8–10 min, 98–2% B; 10–12 min, 2% B.

Mass spectrometric detection of metabolites was performed on QE HF-X (Thermo Fisher, USA) with an ESI ion source. Simultaneous MS1 and MS/MS (Full MS-ddMS2 mode, data-dependent MS/MS) acquisition was used. The parameters were set as follows: spray voltage, 3.60 and −3.60 kV for ESI (+) and ESI (−), respectively; the heated capillary temperature was 325 °C; secondary collision energy (NCE): 20, 30, 40; Top $N = 10$; Scan range: 70–1050 m/z.

Data were collected and processed using Compound Discoverer software (version 3.3). The parameters were set as follows: alignment model: adaptive curve; maximun shift: 0.5 min; mass tolerance: 10 ppm; intensity tolerance: 30%; S/N threshold: 1.5; mass tolerance: 10 ppm; match factor threshold: 10. Normalize the sample to the median of the maximum peak area of all samples. Compounds were identified based on MS2 information using automatic multi-database and spectrum library search tools (including mzCloud, Chemspider, KEGG, and HMDB) and local database search tools (such as mzVault). The variable importance in projection (VIP) value >1 was combined with average ratio-fold change >1.5, $p$ value <0.05, to determine if the metabolite feature was considered statistically significant.

## Crystallization and structural analysis

Crystallization screens were performed using the hanging-drop vapor diffusion method at 16 °C, with drops containing 0.5 uL of the protein solution mixed with 0.5 uL of reservoir solution. Diffraction quality of SA_{CpfC} crystals was obtained in 0.2 M Calcium chloride dihydrate; 20% w/v polyethylene glycol 3350. Crystals were harvested and flash-frozen in liquid nitrogen with 20% glycerol as a cryoprotectant. Complete X-ray diffraction datasets were collected at the BL-18U1 beamline of Shanghai Synchrotron Radiation Facility (SSRF). Diffraction images were processed with the HKL-3000 program. Crystal structures were solved by molecular replacement (MR) using Phaser-MR to obtain the model of AlphaFold. Model building and crystallographic refinement were carried out in Coot v0.9.8.7 and PHENIX v1.18.2. The interactions were analyzed with PyMOL (http://www.pymol.org/) and PDBsum. The figures were generated in PyMOL. Detailed data collection and refinement statistics are listed in Appendix Table S2.

## Molecular docking

The SA_{CpfC} crystal structure (PDB ID: 9VI5), determined in this study, and its Y12A, H181A, and E263A mutants were prepared using the Protein Preparation Wizard within Schrödinger (version 2018-1). This preparation included adding hydrogen atoms, assigning bond orders, and optimizing the hydrogen-bonding network. Molecular docking of SC-43 into the heme-binding site of SA_{CpfC} was performed using the Glide module (Schrödinger, version 2018-1) in extra precision (XP) mode. Grid generation centered on the heme-binding site, with default settings for van der Waals scaling and ligand flexibility. The top-ranked docking pose, based on the Glide XP score, was selected for subsequent MD simulation.

## MD simulation

MD simulation was conducted using the Desmond 2022 academic version with the OPLS3 force field. The SC-43-SA_{CpfC} complex was solvated in an orthorhombic box of SPC water molecules, with a 10 Å buffer distance from the protein surface. The system was neutralized with Na⁺ and Cl⁻ counterions, and additional Na⁺ and Cl⁻ ions were added to achieve a concentration of 0.15 M. Energy minimization was performed using a hybrid steepest descent and limited-memory Broyden-Fletcher-Goldfarb-Shanno (LBFGS) algorithm, followed by equilibration under NPT conditions (300 K, 1.01325 bar) with the Berendsen thermostat and barostat. A 200 ns production run was then carried out with a 2 fs timestep,

employing the RESPA integrator. Trajectory frames were saved every 100 ps. System stability was assessed by calculating the root mean square deviation (RMSD) of the protein-ligand complex heavy atoms relative to the initial frame, with equilibrium achieved after ~60 ns. Consequently, the analysis focused on the 60–200 ns segment of the trajectory.

## Interaction and structural analysis

Intermolecular interactions were quantified using Schrödinger's Simulation Interaction Diagram tool. Hydrogen bonds and π-π stacking interactions between SC-43 and key $SA_{CpfC}$ residues (Tyr12, Glu263, Tyr25, and His181) were monitored throughout the 60–200 ns trajectory, with occupancy percentages calculated relative to the total analyzed frames. For visualization, trajectory clustering was performed using Desmond's trajectory clustering tool, based on the RMSD of the ligand and binding site residues (cutoff: 2.0 Å). The representative structure from the largest cluster was extracted to generate the three-dimensional binding mode depiction. Two-dimensional interaction diagrams were generated using Schrödinger's Ligand Interaction Diagram tool, and interaction frequencies were plotted. All structural visualizations and RMSD plots were generated using PyMOL (version 2.5, open source) and Schrödinger's Maestro interface.

## The activity of SC-43 against MRSA in vivo

The antibacterial activity of SC-43 against MRSA was evaluated using a mouse skin infection model. Female BALB/c mice, aged 6–8 weeks, were randomly divided into four groups ($n = 20$ per group). The infection and treatment group were inoculated intradermally with $10^7$ CFUs of MRSA 252, while the control group received an equivalent volume of sterile PBS. The infection was repeated once, 12 h later. The treatment group was administered SC-43 (0.1 mg/kg) or mupirocin (0.1 mg/kg) via cutaneous administration method 24 h post-infection, followed by once-daily dosing for 3 d (Fig. 6A). The vehicle control group received the same volume as the solvent. The cutaneous injuries were recorded and observed continuously, and the local bacterial load was compared on the 1st, 3rd, 5th, and 7th days, respectively. Compare the size of the original wound with that of the wound on the 7th day to evaluate the rate of wound healing. Sample collection, processing, and CFUs counting were performed according to the previously published protocol (Zeng et al, 2023). The experiments of this study were carried out independently.

## Ethics

Mice were housed under a standard light/dark cycle with food and water ad libitum. All animal procedures were conducted according to the National Institutes of Health Guide for the Care and Use of Laboratory Animals. The Animal Ethical and Experimental Committee of the Army Military Medical University approved the animal experiments (License 2011-04).

## Statistical analysis

Statistical analysis was performed using GraphPad Prism v10. All experiments were performed in biological repetition. The $t$-test and one-way or two-way ANOVA are used to conduct quantitative data analysis for two groups and more than two groups, respectively. $P < 0.05$ was considered statistically significant. $^{****}P < 0.0001$, $^{***}P < 0.001$, $^{**}P < 0.01$, and $^{*}P < 0.05$ as indicated in figure legends.

# Peer review information

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

### The paper explained

**Problem**

Methicillin-resistant *Staphylococcus aureus* (MRSA) is the high-priority pathogen category of the World Health Organization (WHO) bacterial priority pathogens list. Due to the limitations of existing drugs and the emergence of drug resistance, it is urgent to develop new anti-MRSA drugs.

**Results**

We conducted a screening of a phase 1–3 clinical compound library based on antibacterial phenotypes and found that SC-43 exhibited activity against MRSA. The mechanism of action studies showed that SC-43 exerts its antibacterial effect by inhibiting the function of coproporphyrin ferrochelatase (CpfC), thereby interfering with the biosynthesis of heme, ATP levels and causing hyperpolarization in *S. aureus*. We also determined the structure of CpfC derived from *S. aureus* ($SA_{CpfC}$) and revealed the anti-MRSA mechanism of SC-43 at the molecular level.

**Impact**

Our study extends the understanding of the pharmacological activities of SC-43 and revalidates a proof-of-concept for the feasibility of targeting key enzymes such as CpfC in the coproporphyrin-dependent (CPD) heme synthesis pathway for the development of new drugs against Gram-positive bacteria, including MRSA.

Chastre J, Blasi F, Masterton RG, Rello J, Torres A, Welte T (2014) European perspective and update on the management of nosocomial pneumonia due to methicillin-resistant *Staphylococcus aureus* after more than 10 years of experience with linezolid. Clin Microbiol Infect 20:19–36

Cheng MJ, Wu YY, Zeng H, Zhang TH, Hu YX, Liu SY, Cui RQ, Hu CX, Zou QM, Li CC et al (2024) Asymmetric total synthesis of polycyclic xanthenes and discovery of a WalK activator active against MRSA. Nat Commun 15:5879

Clinical and Laboratory Standards Institute [CLSI] (2024) M100 performance standards for antimicrobial susceptibility testing. Clinical and Laboratory Standards Institute

Dailey HA, Gerdes S, Dailey TA, Burch JS, Phillips JD (2015) Noncanonical coproporphyrin-dependent bacterial heme biosynthesis pathway that does not use protoporphyrin. Proc Natl Acad Sci USA 112:2210–2215

Dailey HA, Medlock AE (2022) A primer on heme biosynthesis. Biol Chem 403:985–1003

European Antimicrobial Resistance Collaborators (2022) The burden of bacterial antimicrobial resistance in the WHO European region in 2019: a cross-country systematic analysis. Lancet Public Health 7:e897–e913

Franken H, Mathieson T, Childs D, Sweetman GM, Werner T, Tögel I, Doce C, Gade S, Bantscheff M, Drewes G et al (2015) Thermal proteome profiling for unbiased identification of direct and indirect drug targets using multiplexed quantitative mass spectrometry. Nat Protoc 10:1567–1593

Hammer ND, Reniere ML, Cassat JE, Zhang Y, Hirsch AO, Indriati Hood M, Skaar EP (2013) Two heme-dependent terminal oxidases power *Staphylococcus aureus* organ-specific colonization of the vertebrate host. mBio 4:e00241-13

Hao GF, Zuo Y, Yang SG, Yang GF (2011) Protoporphyrinogen oxidase inhibitor: an ideal target for herbicide discovery. Chimia 65:961–969

Hobbs C, Reid JD, Shepherd M (2017) The coproporphyrin ferrochelatase of *Staphylococcus aureus*: mechanistic insights into a regulatory iron-binding site. Biochem J 474:3513–3522

Hoffmann A, Grubwieser P, Bumann D, Haschka D, Weiss G (2025) Tackling microbial iron homeostasis: novel antibacterial strategies. Trends Pharmacol Sci 46:1004–1017

Hu MH, Chen LJ, Chen YL, Tsai MS, Shiau CW, Chao TI, Liu CY, Kao JH, Chen KF (2017) Targeting SHP-1-STAT3 signaling: a promising therapeutic approach for the treatment of cholangiocarcinoma. Oncotarget 8:65077–65089

Jackson LK, Dailey TA, Anderle B, Warren MJ, Bergonia HA, Dailey HA, Phillips JD (2023) Exploiting differences in heme biosynthesis between bacterial species to screen for novel antimicrobials. Biomolecules 13:1485

Jiang F, Chen Y, Yu J, Zhang F, Liu Q, He L, Musha H, Du J, Wang B, Han P et al (2023) Repurposed fenoprofen targeting SaeR attenuates *Staphylococcus aureus* virulence in implant-associated infections. ACS Cent Sci 9:1354–1373

Layer G (2021) Heme biosynthesis in prokaryotes. Biochim Biophys Acta Mol Cell Res 1868:118861

Lee EY, Caffrey AR (2017) Thrombocytopenia with tedizolid and linezolid. Antimicrob Agents Chemother 62:e01453-17

Li Y, Ma H (2024) Drug repurposing: insights into the antimicrobial effects of AKBA against MRSA. AMB Express 14:5

Liu L, Martínez JL, Liu Z, Petranovic D, Nielsen J (2014) Balanced globin protein expression and heme biosynthesis improve production of human hemoglobin in *Saccharomyces cerevisiae*. Metab Eng 21:9–16

Liu Q, He D, Wang L, Wu Y, Liu X, Yang Y, Chen Z, Dong Z, Luo Y, Song Y (2024) Efficacy and safety of antibiotics in the treatment of methicillin-resistant *Staphylococcus aureus* (MRSA) infections: a systematic review and network meta-analysis. Antibiotics 13:866

Lobo SA, Scott A, Videira MA, Winpenny D, Gardner M, Palmer MJ, Schroeder S, Lawrence AD, Parkinson T, Warren MJ et al (2015) Staphylococcus aureus haem biosynthesis: characterisation of the enzymes involved in final steps of the pathway. Mol Microbiol 97:472–487

Mandell LA, Wunderink RG, Anzueto A, Bartlett JG, Campbell GD, Dean NC, Dowell SF, File TM Jr, Musher DM, Niederman MS et al (2007) Infectious Diseases Society of America/American Thoracic Society consensus guidelines on the management of community-acquired pneumonia in adults. Clin Infect Dis 44:S27–S72

Meng Q, Wang X, Huang X, Li C, Yu Z, Li P, Liu X, Wen Z (2024) Repurposing benzbromarone as an antibacterial agent against Gram-positive bacteria. ACS Infect Dis 10:4208–4221

Nazli A, Tao W, You H, He X, He Y (2024) Treatment of MRSA infection: where are we? Curr Med Chem 31:4425–4460

Perez-Riverol Y, Bandla C, Kundu DJ, Kamatchinathan S, Bai J, Hewapathirana S, John NS, Prakash A, Walzer M, Wang S et al (2025) The PRIDE database at 20 years: 2025 update. Nucleic Acids Res 53:D543–D553

Poulos TL (2014) Heme enzyme structure and function. Chem Rev 114:3919–3962

Savitski MM, Reinhard FB, Franken H, Werner T, Savitski MF, Eberhard D, Martinez Molina D, Jafari R, Dovega RB, Klaeger S et al (2014) Tracking cancer drugs in living cells by thermal profiling of the proteome. Science 346:1255784

Sishtla K, Lambert-Cheatham N, Lee B, Han DH, Park J, Sardar Pasha SPB, Lee S, Kwon S, Muniyandi A, Park B et al (2022) Small-molecule inhibitors of ferrochelatase are antiangiogenic agents. Cell Chem Biol 29:1010–1023

Su TH, Shiau CW, Jao P, Yang NJ, Tai WT, Liu CJ, Tseng TC, Yang HC, Liu CH, Huang KW et al (2017) Src-homology protein tyrosine phosphatase-1 agonist, SC-43, reduces liver fibrosis. Sci Rep 7:1728

Swift SM, Sauve K, Cassino C, Schuch R (2021) Exebacase Is Active In vitro in pulmonary surfactant and is efficacious alone and synergistic with daptomycin in a mouse model of lethal *Staphylococcus aureus* lung infection. Antimicrob Agents Chemother 65:e0272320

Videira MAM, Lobo SAL, Silva LSO, Palmer DJ, Warren MJ, Prieto M, Coutinho A, Sousa FL, Fernandes F, Saraiva LM (2018) *Staphylococcus aureus* haem biosynthesis and acquisition pathways are linked through haem monooxygenase IsdG. Mol Microbiol 109:385–400

Videira MAM, Lobo SAL, Sousa FL, Saraiva LM (2020) Identification of the sirohaem biosynthesis pathway in *Staphylococcus aureus*. FEBS J 287:1537–1553

Wang C, Ji Y, Huo X, Li X, Lu W, Zhang Z, Dong W, Wang X, Chen H, Tan C (2024) Discovery of salifungin as a repurposed antibiotic against methicillin-resistant *Staphylococcus aureus* with limited resistance development. ACS Infect Dis 10:1576–1589

WHO (2024) WHO Bacterial Priority Pathogens List, 2024: bacterial pathogens of public health importance to guide research, development and strategies to prevent and control antimicrobial resistance. World Health Organization

Wu R, Wu Y, Wu P, Li H, She P (2024) Bactericidal and anti-quorum sensing activity of repurposing drug visomitin against *Staphylococcus aureus*. Virulence 15:2415952

Yang H, Zhang Y, Feng X, An Z (2023) Bleeding complications in vancomycin-induced thrombocytopenia: a real-world postmarketing pharmacovigilance analysis. Clin Ther 45:868–872

Zeng H, Cheng M, Liu J, Hu C, Lin S, Cui R, Li H, Ye W, Wang L, Huang W (2023) Pyrimirhodomyrtone inhibits *Staphylococcus aureus* by affecting the activity of NagA. Biochem Pharmacol 210:115455

Zhang L, Zhang Y, Tian L, Shen Q, Ma X (2024) Doxifluridine effectively kills antibiotic-resistant *Staphylococcus aureus* in chronic obstructive pulmonary disease. Microbiol Spectr 12:e0180524

## Acknowledgements

We thank M. Sun from the Shenzhen Bay Laboratory for the SPR data Analysis and the support of Beijing Qinglian Biotech Co., Ltd for the TPP data. This work was supported by the National Natural Science Foundation of China (82173859 to WH); Guangdong Provincial Clinical Research Center for Laboratory Medicine

(2023B110008 to XY); Shenzhen Science and Technology and Innovation Commission (GJHZ20220913142800001 and SYSPG20241211173920041 to WH, ZDSYS20111002 to WH and XY); Shenzhen Clinical Research Center for Respiratory Disease (LCYSSQ20220823091203007 to WH); Shenzhen People's Hospital Physician Scientist Training "Five Three Program" (SYWGSCGZH202401 to WH). Fundamental Research Funds for the Central Universities (No. 24qnpy061 to XL).

## Author contributions

**Yini Huang**: Software; Investigation; Visualization; Methodology. **Yan Ye**: Investigation; Methodology. **Xinmei Zhu**: Investigation; Methodology. **Dengpan Liang**: Investigation; Methodology. **Ruiqin Cui**: Data curation; Validation. **Xiaopeng Yuan**: Resources; Project administration. **Xitao Li**: Resources; Software; Investigation; Visualization. **Quanming Zou**: Resources; Project administration. **Haibo Li**: Resources; Supervision; Project administration. **Wei Huang**: Conceptualization; Resources; Supervision; Funding acquisition; Writing—original draft; Writing—review and editing.

Source data underlying figure panels in this paper may have individual authorship assigned. Where available, figure panel/source data authorship is listed in the following database record: biostudies:S-SCDT-10_1038-S44321-026-00418-4.

## Disclosure and competing interests statement

The authors declare no competing interests.

