## [Peer Review File · EMBO Molecular Medicine]

SHP-1 agonist SC-43 limits Methicillin-resistant *Staphylococcus aureus* infection through inhibition of heme biosynthesis

Yini Huang, Yan Ye, Xinmei Zhu, Dengpan Liang, Ruiqin Cui, Xiaopeng Yuan, Xitao Li, Quanming Zou, Haibo Li, and Wei Huang

Corresponding authors: Wei Huang (weihuang@mail.sustech.edu.cn) , Xitao Li (lixt78@mail.sysu.edu.cn), Haibo Li (lihaibo@tmmu.edu.cn), Quanming Zou (qmzou@tmmu.edu.cn)

Review Timeline:

Submission Date:	20th Sep 25
Editorial Decision:	21st Nov 25
Revision Received:	22nd Jan 26
Editorial Decision:	20th Feb 26
Revision Received:	24th Feb 26
Accepted:	16th Mar 26

Editor: Zeljko Durdevic

Transaction Report:

21st Nov 2025

Dear Dr. Huang,

Thank you for the submission of your manuscript to EMBO Molecular Medicine, and please accept my apologies for the unusual delay in getting back to you. We have now received feedback from two of the three reviewers who agreed to evaluate your manuscript. As the referee #2 will unfortunately not be able to return his/her report in a timely manner, we prefer to make a decision now in order to avoid further delay in the process. Should referee #2 provide a report, we will send it to you, with the understanding that we will not ask for an additional revision.

As you will see from their reports pasted below, both referees recognize potential interest of the manuscript but also raise serious concerns that should be addressed in a major revision. If you would like to discuss further the points raised by the referees, I am available to do so via email or video. Let me know if you are interested in this option.

We would welcome the submission of a revised version within three months for further consideration. Please let us know if you require longer to complete the revision.

I look forward to receiving your revised manuscript.

Yours sincerely,

Zeljko Durdevic

Zeljko Durdevic
Senior Editor
EMBO Molecular Medicine

When preparing your revised manuscript, please refer to our guidelines: <https://link.springer.com/journal/44321/submission-guidelines#cms-Revised-submissions>. We perform an initial quality control of all revised manuscripts before re-review; failure to include requested items will delay the evaluation of your revision.

We require:

- 1) A .docx formatted version of the manuscript text (including legends for main figures, EV figures and tables). Please make sure that the changes are highlighted to be clearly visible.
- 2) Individual production quality figure files as .eps, .tif, .jpg (one file per figure). For guidance, download the 'Figure Guide PDF': <https://media.springernature.com/original/springer-cms/rest/v1/content/27825798/data/v1>.
- 3) A .docx formatted letter INCLUDING the reviewers' reports and your detailed point-by-point responses to their comments. As part of the EMBO Press transparent editorial process, the point-by-point response is part of the Review Process File (RPF), which will be published alongside your paper.
- 4) A complete author checklist, which you can download from our author guidelines. Please insert information in the checklist that is also reflected in the manuscript. The completed author checklist will also be part of the RPF.
- 5) Please note that all corresponding authors are required to supply an ORCID ID for their name upon submission of a revised

manuscript.

6) It is mandatory to include a 'Data Availability' section after the Materials and Methods. Before submitting your revision, primary datasets produced in this study need to be deposited in an appropriate public database, and the accession numbers and database listed under 'Data Availability'. Please remember to provide a reviewer password if the datasets are not yet public.

7) For data quantification: please specify the name of the statistical test used to generate error bars and P values, the number (n) of independent experiments (specify technical or biological replicates) underlying each data point and the test used to calculate p-values in each figure legend. The figure legends should contain a basic description of n, P and the test applied. Graphs must include a description of the bars and the error bars (s.d., s.e.m.).

9) Our journal encourages inclusion of *data citations in the reference list* to directly cite datasets that were re-used and obtained from public databases. Data citations in the article text are distinct from normal bibliographical citations and should directly link to the database records from which the data can be accessed. In the main text, data citations are formatted as follows: "Data ref: Smith et al, 2001" or "Data ref: NCBI Sequence Read Archive PRJNA342805, 2017". In the Reference list, data citations must be labeled with "[DATASET]". A data reference must provide the database name, accession number/identifiers and a resolvable link to the landing page from which the data can be accessed at the end of the reference.

- the medical issue you are addressing,

- the results obtained and

- their clinical impact.

12) Author contributions: You will be asked to provide CRediT (Contributor Role Taxonomy) terms in the submission system. These replace a narrative author contribution section in the manuscript.

13) A Conflict of Interest statement should be provided in the main text.

14) Every published paper includes a 'Synopsis' to further enhance discoverability. Synopses are displayed on the journal webpage and are freely accessible to all readers. They include a short stand first (maximum of 300 characters, including space) as well as 2-5 one-sentences bullet points that summarizes the paper. Please write the bullet points to summarize the key NEW findings. They should be designed to be complementary to the abstract - i.e. not repeat the same text. We encourage inclusion of key acronyms and quantitative information (maximum of 30 words / bullet point). Please use the passive voice. Please attach these in a separate file or send them by email, we will incorporate them accordingly.

15) Include a Reagents and Tools Table as part of the Methods section, which can be downloaded from our author guidelines.

***** Reviewer's comments *****

Referee #1 (Comments on Novelty/Model System for Author):

The authors may use another independent mouse model to evaluate the drug's effect on MRSA infection/colonization.

Referee #1 (Remarks for Author):

In this study, Huang et al demonstrate that SC-43, a clinical-stage SHP-1 antagonist, exerts potent bactericidal activity against MRSA and other Gram-positive pathogens. The authors claim that SC-43 inhibits CpfC, a key enzyme in heme biosynthesis pathway unique to Gram-positive bacteria. Overall, the experiments are convincing and well controlled, and the claims are mostly supported. The findings will likely appeal to a broad audience. However, there are some concerns which should be addressed prior to publication.

Major Concerns:

- 1) Since SC-43 is an orally active Sorafenib derivative with potent anti-fibrotic and anti-cancer effects, please elaborate on the rationale for linking SC-43 with MRSA. What was the initial hypothesis or screening strategy that led to this discovery?
- 2) Lines 120-123: Were other candidate proteins from the TPP experiment validated to confirm the reliability of the results? Furthermore, among the identified candidates, aldolase is a conserved and crucial glycolytic enzyme. If SC-43 also affects its function, how can the specificity of SC-43 for the pathogen (via CpfC) be guaranteed, considering potential off-target effects in bacterial metabolism?
- 3) Does SC-43 affect other metabolic pathways in *Staphylococcus aureus*/ Gram-positive bacteria beyond heme biosynthesis? A broader analysis of metabolic consequences would strengthen the claim of mechanism-specific action.
- 4) Do the Y12, H181, and E263 point mutations in CpfC affect intracellular heme synthesis and ATP levels in bacteria?
- 5) I notice that the animal model citation (reference 24) is also from the authors' previous work. Are there data from other, independent mouse models to evaluate the drug's effect on MRSA infection/colonization?
- 6) Does SC-43 affect the normal skin microbiota (e.g., *Staphylococcus epidermidis*)? Given the abundance of commensal Gram-positive bacteria on the skin, could SC-43 disrupt the normal microbiota?
- 7) The authors propose that the decreased intracellular heme level primarily affects bacterial respiration and oxidative stress responses as a cofactor for various enzymes. However, heme also serves as an essential iron source for bacterial growth, among other potential functions. The authors have not ruled out these additional potential roles of heme in the observed phenotype. This issue should be discussed.

Minor Concerns:

- 1) Table S1 is confusing and lacks sufficient annotation and description for easy interpretation.
- 2) Figure 5A uses 20 mice per group but lacks statistical analysis.
- 3) Figure 1 D and E lack appropriate negative controls.

Referee #3 (Remarks for Author):

Comment on "SC-43 Inhibits Methicillin-resistant *Staphylococcus aureus* Through Inhibition of Heme Biosynthesis"
Sorafenib is a multi-kinase inhibitor approved for the treatment of hepatocellular carcinoma (HCC). The sorafenib derivative SC-43 also has antitumor activity and ameliorates liver fibrosis through upregulation of the phosphatase SHP-1. This paper shows that SC-43 additionally has antimicrobial activity against Gram-positive bacteria, including MRSA. In *S. aureus*, SC-43 inactivates the key enzyme coproporphyrin ferrochelatase (CpfC) of the coproporphyrin-dependent (CPD) heme-synthesis pathway.

The authors solved the crystal structure of the *S. aureus* CpfC (SaCpfC) and processed it using the Protein Preparation Wizard (Schrödinger). SC-43 was docked into the heme-binding site using Glide XP. To verify the pocket, Y12A, H181A and E263A SaCpfC mutants were generated; all showed reduced binding to SC-43. In an MRSA murine skin wound infection model, SC-43 treatment significantly reduced wound size and local bacterial load. Exogenous heme supplementation abolished the inhibitory effect of SC-43.

Overall, the authors convincingly demonstrate that the repurposed drug SC-43 - known for antitumor activity - also has antimicrobial activity. A major strength is the identification of coproporphyrin ferrochelatase (CpfC) from the CPD heme-synthesis

pathway as the molecular target, together with mapping of a proposed binding site. A limitation is that the activity of SC-43 is abolished by exogenous heme, which could restrict clinical application.

The manuscript is well written and the experiments are logical and carefully executed. I have several questions and suggestions for the authors (below) that, if addressed, would strengthen the mechanistic claims and place the findings in context.

There are some questions:

Do SHP-1 or STAT3 share sequence or structural similarity with CpfC, or whether their active/binding sites are structurally similar (heme pocket mimicry). Please provide either (a) a short BLAST/structure-based comparison showing lack/presence of homology, or (b) an argument why off-target binding to human SHP-1/STAT3 is unlikely (e.g., very different folds / lack of heme pocket). If feasible, include percent identity and structural alignment figures or a brief statement in the Discussion.

What were the initial indications that CpfC is the cellular target? In the paper it should be mentioned the lead observations that led to CpfC identification (e.g., genetic resistance selection, metabolomics showing accumulation of coproporphyrin precursors, affinity capture/pull-down, phenotypic rescue by heme). If this was a chemical genetics screen or unbiased proteomics result, please summarize the key experiments and data that prioritized CpfC.

For the murine wound model: indicate dosing schedule and route. Discuss the implication of heme in serum/tissue for efficacy in systemic vs. topical applications.

Docking is useful, but co-crystallization of SC-43 with SaCpfC (or soaking) would be much stronger evidence of the binding mode. If a co-crystal was not obtained, state this limitation. At minimum, provide docking score, major interactions, and whether the predicted contacts agree with the mutant binding data.

The hem biosynthesis has been very well studied in the group of Saraiva and they also identified hem-biosynthesis inhibitors. The papers should be discussed.

Lobo, S.A., Scott, A., Videira, M.A., Winpenny, D., Gardner, M., Palmer, M.J., Schroeder, S., Lawrence, A.D., Parkinson, T., Warren, M.J., and Saraiva, L.M. (2015) Staphylococcus aureus haem biosynthesis: characterisation of the enzymes involved in final steps of the pathway. *Mol Microbiol* 97: 472-487.

Mendes, S.S., Marques, J., Mesterhazy, E., Straetener, J., Arts, M., Pissarro, T., Reginold, J., Berscheid, A., Bornikoel, J., Kluj, R.M., Mayer, C., Oesterhelt, F., Friaes, S., Royo, B., Schneider, T., Brotz-Oesterhelt, H., Romao, C.C., and Saraiva, L.M. (2022) Synergetic Antimicrobial Activity and Mechanism of Clotrimazole-Linked CO-Releasing Molecules. *ACS Bio Med Chem Au* 2: 419-436.

Videira, M.A.M., Lobo, S.A.L., Silva, L.S.O., Palmer, D.J., Warren, M.J., Prieto, M., Coutinho, A., Sousa, F.L., Fernandes, F., and Saraiva, L.M. (2018) Staphylococcus aureus haem biosynthesis and acquisition pathways are linked through haem monooxygenase IsdG. *Mol Microbiol* 109: 385-400.

Videira, M.A.M., Lobo, S.A.L., Sousa, F.L., and Saraiva, L.M. (2020) Identification of the sirohaem biosynthesis pathway in Staphylococcus aureus. *FEBS J* 287: 1537-1553.

Referee #1 (Comments on Novelty/Model System for Author)

The authors may use another independent mouse model to evaluate the drug's effect on MRSA infection/colonization.

Response: Thank you very much for the reviewers' comments. We are deeply sorry for the confusion caused by the issues described in the manuscript. Our intention was to indicate that the construction process of the animal experiment was according to our previously published protocol only. The animal experiments in this manuscript and those in the previously published papers were conducted at two different times, and the data were completely independent. To prevent readers from having such confusion, we have revised the wording in the revised manuscript, please see lines 883 to 884 of the revised manuscript.

Referee #1 (Remarks for Author)

In this study, Huang et al demonstrate that SC-43, a clinical-stage SHP-1 antagonist, exerts potent bactericidal activity against MRSA and other Gram-positive pathogens. The authors claim that SC-43 inhibits CpfC, a key enzyme in heme biosynthesis pathway unique to Gram-positive bacteria. Overall, the experiments are convincing and well controlled, and the claims are mostly supported. The findings will likely appeal to a broad audience. However, there are some concerns which should be addressed prior to publication.

Major Concerns:

1) Since SC-43 is an orally active Sorafenib derivative with potent anti-fibrotic and anti-cancer effects, please elaborate on the rationale for linking SC-43 with MRSA. What was the initial hypothesis or screening strategy that led to this discovery?

Response: Thank you very much for the reviewers' comments. Indeed, we overlooked the description of the process of discovering the antibacterial activity of SC-43. In the "MATERIALS AND PROTOCOLS" section of the revised manuscript, we have added a "screening strategy" section. Please see Appendix Table S1 and lines 91 to 94 and 597 to 600 of the revised manuscript.

2) Lines 120-123: Were other candidate proteins from the TPP experiment validated to confirm the reliability of the results? Furthermore, among the identified candidates, aldolase is a conserved and crucial glycolytic enzyme. If SC-43 also affects its function, how can the specificity of SC-43 for the pathogen (via CpfC) be guaranteed, considering potential off-target effects in bacterial metabolism?

Response: Thank you very much for the reviewers' comments. We mainly rely on the antibacterial

phenotypes and antibacterial spectra (Effective against Gram-positive bacteria used in this study but ineffective against Gram-negative bacteria, even when used in combination with outer membrane disruptors, please see Table 1 of the revised manuscript) to make judgments on whether the candidate proteins with statistical significance from the TPP experiment are potential targets. Therefore, two conditions need to be met for a protein to be considered a potential target. Firstly, it must be a necessary protein for Gram-positive bacteria used in this study. Secondly, it is very likely to be non-essential for Gram-negative bacteria (the possibility that SC-43 is ineffective against Gram-negative bacteria due to the obstruction of the outer membrane was ruled out by using SC-43 with outer membrane disruptor). Therefore, the proteins that do not meet these two conditions were excluded from our study for mechanism studies.

Regarding aldolase, we fully agree with the reviewer's opinion that aldolase is a conserved and crucial glycolytic enzyme. The reason why we ruled out it as the target is that in *Staphylococcus aureus*, there are proteins that can compensate for the function of aldolase. In our study, the aldolase identified through TPP is Fructose-bisphosphate aldolase class I. In *Staphylococcus aureus*, there is also Fructose-bisphosphate aldolase class II (Biochemistry. 2014;53(48):7604-14.). Therefore, the presence of class II aldolase makes it highly unlikely that the identified fructose-bisphosphate aldolase in this study is the target responsible for the antibacterial effect (please see lines 500 to 518 of the revised manuscript).

To clarify the effects of SC-43 on other metabolic pathways, we conducted additional metabolomics experiments, which revealed that in addition to the porphyrin metabolism pathway closely related to CpfC, the O-antigen nucleotide sugar biosynthesis and lysine degradation pathways also be affected by SC-43. Therefore, we proposed that SC-43 has multiple effects on bacteria, please see lines 513 to 518 and Fig. 3A of the revised manuscript.

3) Does SC-43 affect other metabolic pathways in *Staphylococcus aureus*/ Gram-positive bacteria beyond heme biosynthesis? A broader analysis of metabolic consequences would strengthen the claim of mechanism-specific action.

Response: Thank you very much for the reviewers' comments. To clarify the effects of SC-43 on other metabolic pathways, we conducted additional metabolomics experiments, which revealed that in addition to the porphyrin metabolism pathway closely related to CpfC, the O-antigen nucleotide sugar biosynthesis and lysine degradation pathways also be affected by SC-43. Therefore, we proposed that SC-43 has multiple effects on bacteria, please see lines 513 to 518 and Fig. 3A of the revised manuscript.

4) Do the Y12, H181, and E263 point mutations in CpfC affect intracellular heme synthesis and ATP levels in bacteria?

Response: Thank you very much for the reviewers' comments. In the revised manuscript, we conducted experiments to measure the enzymatic activity of the proteins with Y21A, H181A or E263A point mutations. As shown in Fig. 1E, we were unable to detect the catalytic function of

iron chelation for the Y12A, H181A and E263A mutant proteins (Fig. 1E).

In addition, through standard passaging experiment, no spontaneous mutant strain resistant to SC-43 was obtained. This indicates that bacteria are not prone to developing resistance to it, and at the same time, it also hinders our acquisition of genetic evidence and prevents us from evaluating the heme synthesis and ATP levels in bacteria with Y12, H181, and E263 point mutations in CpfC. Given that the mutations at these sites have a significant impact on the enzymatic activity of CpfC (Fig. 1E), this might explain why spontaneous drug-resistant strains and genetic evidence could not be obtained, please see lines 488 to 498 and Fig. 1E of the revised manuscript.

5) I notice that the animal model citation (reference 24) is also from the authors' previous work. Are there data from other, independent mouse models to evaluate the drug's effect on MRSA infection/colonization?

Response: We are deeply sorry for the confusion caused by the issues described in the manuscript. Our intention was to indicate that the construction process of the animal experiment was according to our previously published protocol, but it does not mean that these two experiments were conducted simultaneously. The animal experiment in this manuscript and those in the previously published papers were conducted at two different times, and the data were completely independent with no shared data. To prevent readers from having such confusion, we have revised the wording in the revised manuscript, please see lines 880 to 884 of the revised manuscript.

6) Does SC-43 affect the normal skin microbiota (e.g., *Staphylococcus epidermidis*)? Given the abundance of commensal Gram-positive bacteria on the skin, could SC-43 disrupt the normal microbiota?

Response: Thank you very much for the reviewers' comments. We have added an AST test for *Staphylococcus epidermidis* in the revised manuscript. The results show that SC-43 also has antibacterial activity against *S. epidermidis* (Table 1). Therefore, we propose that the resulting impact on the normal skin flora cannot be ignored, please see lines 533 to 535 of the revised manuscript.

7) The authors propose that the decreased intracellular heme level primarily affects bacterial respiration and oxidative stress responses as a cofactor for various enzymes. However, heme also serves as an essential iron source for bacterial growth, among other potential functions. The authors have not ruled out these additional potential roles of heme in the observed phenotype. This issue should be discussed.

Response: Thank you very much for the reviewers' comments. Indeed, apart from serving as a cofactor for key enzymes involved in bacterial respiration and oxidative stress responses, many other biological functions of heme may also be related to the phenotypes we observed. We have elaborated and discussed this in the discussion section of the revised manuscript. Please see lines 479 to 482 of the revised version.

Minor Concerns:

1) Table S1 is confusing and lacks sufficient annotation and description for easy interpretation.

Response: Thank you very much for the reviewers' comments. We have simplified the content of Table S1 by eliminating some irrelevant indicators and added annotations to the table. Please see Appendix Table S2 of the revised version.

2) Figure 5A uses 20 mice per group but lacks statistical analysis.

Response: Thank you very much for the reviewers' comments. We have compared the size of the original wound with that of the wound on the 7th day to evaluate the rate of wound healing, please see figure 6C of the revised manuscript.

3) Figure 1 D and E lack appropriate negative controls.

Response: Thank you very much for the reviewers' comments. We have added vancomycin as a negative control in fig 1D and E, and no binding of vancomycin to CpfC or its effect on enzyme activity was found.

Referee #2

Remarks for Author:

This manuscript shows that SC-43, a small molecule previously developed as an SHP-1 modulator with anti-tumor/anti-fibrotic activity, also has potent antibacterial activity against Gram-positive pathogens including MRSA and VRE. Thermal proteome profiling identified coproporphyrin ferrochelatase (CpfC) in the coproporphyrin-dependent heme biosynthesis pathway as a putative *S. aureus* target. SC-43 is shown to bind recombinant SaCpfC by SPR, inhibiting its ferrochelatase activity, lowering bound heme and ATP levels, and its growth inhibition was partially rescued by exogenous heme. The authors solve the SaCpfC crystal structure and, using docking and MD simulations plus mutagenesis, propose that SC-43 occupies the heme-binding pocket. In a murine MRSA skin-wound model, topical SC-43 at 0.1 mg/kg reduced lesion size and bacterial burden, outperforming mupirocin. SC-43 is proposed as a repurposing candidate, and selective inhibition of the CPD heme pathway is advanced as an interesting anti-Gram-positive strategy.

The study's main strengths are the identification of a clinically advanced small molecule with potent anti-MRSA activity, evidence linking its activity to the CPD heme pathway via CpfC, a new *S. aureus* CpfC structure, and in vivo efficacy in a murine infection model. However, several gaps remain. The following specific comments need to be addressed before the

manuscript can be further considered.

Specific comments:

• **Mechanistic assignment to CpfC not fully supported. TPP identifies multiple stabilized proteins, but only CpfC is followed up, and there is no genetic validation (e.g. in vivo mutagenesis or overexpression in *S. aureus*). The heme rescue is partial and not fully quantified. I suggest softening some of the claims made.**

Response: Thank you very much for the reviewers' comments. We attempted to isolate the spontaneous drug-resistant strains of bacteria against SC-43, but failed. This indicates that bacteria are not prone to developing resistance to SC-43 on the one hand, and it also prevented us from obtaining genetic evidence on the other hand. Therefore, in the revised manuscript, we added the content about overexpression of *cpfC*. The results showed that after overexpressing of *cpfC*, the minimum inhibitory concentration (MIC) of the bacteria against SC-43 increased by 16-fold times, indicating the correlation between CpfC and the antibacterial activity of SC-43. Please see table 1 and lines 172 to 175 of the revised version.

We fully agree with the reviewer's opinion that the claims made in this study should be softened. The additional metabolomics experiments in revised manuscript also verified that the effects of SC-43 on other metabolic pathways as O-antigen nucleotide sugar biosynthesis and lysine degradation pathways. Therefore, we proposed that SC-43 has multiple effects on bacteria, please see lines 513 to 516 and Fig. 3A of the revised manuscript.

• **Only the apo SaCpfC structure is solved; the proposed SC-43 pose relies on docking/MD. The KD is modest, kinetics are not deeply analyzed, and the mutagenesis only modestly alters affinity without functional (enzyme or MIC) data. This needs clarification and/or a limitations section.**

Response: Thank you very much for the reviewers' comments. In the revised manuscript, we conducted experiments to measure the enzymatic activity of the proteins with Y21A, H181A or E263A point mutations. As shown in Fig. 1E, we were unable to detect the catalytic function of iron chelation for the Y12A, H181A and E263A mutant proteins. Meanwhile, we performed molecular docking studies for SC-43 against the wild-type SA_{CpfC} protein and its mutants (Y12A, H181A, E263A) using the extra precision (XP) mode of Glide in the Schrödinger suite.

As shown in Appendix Fig. S2, the binding affinity, as reflected by the Glide GScore, was weakened for all mutants compared to the wild-type protein (GScore = -5.505). This computational finding is consistent with our experimental Surface Plasmon Resonance (SPR) data (Table 2 in the main text), which also showed reduced affinities for the mutants. The analysis of the binding poses reveals the structural basis for this decrease. In the Y12A mutant complex, the mutation to alanine abolishes both a critical hydrogen bond and a π - π stacking interaction that were observed with the tyrosine residue in the wild-type protein. Similarly, the binding poses for the H181A and E263A mutants show the loss of key interactions present in the wild-type complex. The disruption of these specific interactions provides a plausible explanation for the lower binding

affinities measured by SPR.

However, we fully agree with the reviewer's opinion that the limitation of the lack of functional data (MIC) in this study. We have described in the discussion section, please see lines 488 to 498 of the revised manuscript.

• The CPD vs PPD selectivity argument is largely conceptual; there are no data on human ferrochelatase inhibition or heme-dependent host pathways at antibacterial concentrations. Also, SC-43 is inconsistently described as an SHP-1 agonist vs antagonist.

Response: We completely agree with the reviewer's viewpoint. More experiments are needed to verify the selectivity arguments of CPD vs PPD. At present, we can only say that based on the results of this study, compared with the antibacterial spectrum of SC-43, the inference that CpfC is the antibacterial target just revalidate the argument regarding CPD vs PPD. We have explained this in the revised manuscript, please see lines 540 to 541.

We are deeply sorry for the inconsistency in the use of "agonist" and "antagonist" in the manuscript. We have made corrections in the revised manuscript. We are extremely grateful for the reviewers' comments, which have significantly improved the quality of our manuscript.

• The manuscript concludes that SC-43 is inactive against Gram-negative bacteria and that the outer membrane is not the key barrier, yet MICs drop in the presence of polymyxin B for *P. aeruginosa* and *A. baumannii*. This should be discussed.

Response: Thank you very much for the reviewers' comments. We have discussed the MICs drop in the presence of Polymyxin B nonapeptide for *P. aeruginosa* and *A. baumannii* in the revised manuscript. As can be seen from Table 1 of the revised manuscript, when Polymyxin B nonapeptide was added to disrupt the bacterial outer membrane, the MIC values of SC-43 for *A. baumannii* and *P. aeruginosa* decreased to 12.5 and 3.13 $\mu\text{g/mL}$, respectively. This indicates that there is a synergistic effect between SC-43 and the outer membrane disruptors of Gram-negative bacteria. The results of metabolomics indicate that, in addition to the porphyrin metabolic pathway closely related to CpfC, SC-43 can also affect O-antigen nucleotide sugar biosynthesis and lysine degradation pathways, suggesting its multiple effects on bacteria (Fig. 3A). The MICs drop in the presence of PBNA for *P. aeruginosa* and *A. baumannii* may be related to these metabolic pathways, please see lines 107 to 109 and 513 to 518 of the revised manuscript.

• There are no data on the frequency of resistance or possible resistance mechanisms (e.g. cpfC mutations, efflux). Given the focus on MRSA therapy, at least a preliminary resistance assay/passaging experiment would be important as this is quite standard in the field.

Response: Thank you very much for the reviewers' comments. We completely agree with that resistance assay/passaging experiment would be important as this is quite standard in the field. In fact, we attempted to isolate the spontaneous drug-resistant strains of bacteria by passaging in medium with serially increased concentrations of SC-43, but failed. This indicates that bacteria are

not prone to developing resistance to SC-43 on the one hand, and it also prevented us from obtaining genetic evidence on the other hand.

In this situation, we added the content about overexpression of *cpfC*. The results showed that after overexpressing of *cpfC*, the minimum inhibitory concentration (MIC) of the bacteria against SC-43 increased by 16-fold times, indicating the correlation between CpfC and the antibacterial activity of SC-43. Moreover, we were unable to detect the catalytic function of iron chelation for the Y12A, H181A and E263A mutant proteins, thereby confirming the significance of these amino acid sites (Fig. 1E), this might explain why spontaneous drug-resistant strains and genetic evidence could not be obtained, please see Table 1 and lines 361 to 363 and 488 to 498 of the revised manuscript.

• Only bound heme and ATP were measured. Though perhaps beyond the scope of this paper, it would strengthen the story to assess effects on respiration, membrane potential, or virulence factor production and link these phenotypes to CpfC inhibition.

Response: Thank you very much for the reviewers' comments. We have added the experiment to assess the effect on membrane potential and bacterial metabolism, the results showed that SC-43 can cause hyperpolarization in *S. aureus* and inhibit O-antigen nucleotide sugar biosynthesis, lysine degradation and porphyrin metabolism (Fig. 2D, 3), please see lines 245 to 247 and 272 to 284 of the revised manuscript.

• The paper would benefit from careful language and formatting edits.

Response: Thank you very much for the reviewers' comments. The authors have carefully re-edited the manuscript, especially in the layout of the figures, where we have made revisions. If there are any further omissions, please let us know so that we can make the revisions. We are extremely grateful for the reviewers' comments, which have significantly improved the quality of our manuscript.

Referee #3 (Remarks for Author)

Comment on "SC-43 Inhibits Methicillin-resistant Staphylococcus aureus Through Inhibition of Heme Biosynthesis"

Sorafenib is a multi-kinase inhibitor approved for the treatment of hepatocellular carcinoma (HCC). The sorafenib derivative SC-43 also has antitumor activity and ameliorates liver fibrosis through upregulation of the phosphatase SHP-1. This paper shows that SC-43 additionally has antimicrobial activity against Gram-positive bacteria, including MRSA. In *S. aureus*, SC-43 inactivates the key enzyme coproporphyrin ferrochelatase (CpfC) of the

coproporphyrin-dependent (CPD) heme-synthesis pathway.

The authors solved the crystal structure of the *S. aureus* CpfC (SaCpfC) and processed it using the Protein Preparation Wizard (Schrödinger). SC-43 was docked into the heme-binding site using Glide XP. To verify the pocket, Y12A, H181A and E263A SaCpfC mutants were generated; all showed reduced binding to SC-43. In an MRSA murine skin wound infection model, SC-43 treatment significantly reduced wound size and local bacterial load. Exogenous heme supplementation abolished the inhibitory effect of SC-43.

Overall, the authors convincingly demonstrate that the repurposed drug SC-43 - known for antitumor activity - also has antimicrobial activity. A major strength is the identification of coproporphyrin ferrochelatase (CpfC) from the CPD heme-synthesis pathway as the molecular target, together with mapping of a proposed binding site. A limitation is that the activity of SC-43 is abolished by exogenous heme, which could restrict clinical application.

The manuscript is well written and the experiments are logical and carefully executed. I have several questions and suggestions for the authors (below) that, if addressed, would strengthen the mechanistic claims and place the findings in context.

There are some questions:

Do SHP-1 or STAT3 share sequence or structural similarity with CpfC, or whether their active/binding sites are structurally similar (heme pocket mimicry). Please provide either (a) a short BLAST/structure-based comparison showing lack/presence of homology, or (b) an argument why off-target binding to human SHP-1/STAT3 is unlikely (e.g., very different folds / lack of heme pocket). If feasible, include percent identity and structural alignment figures or a brief statement in the Discussion.

Response: Thank you very much for the reviewers' comments. We completely agree with the reviewer's comments and have aligned the sequences of SHP-1 with CpfC (STAT3 was not the direct target of SC-43 as anti-tumor drug). The sequence alignment result showed that the identity between SA_{CpfC} and human SHP-1 is 16.34% and no conserved amino acids for catalyzing iron chelation were found in the SHP-1 sequence either (Appendix Fig. S2). This indicates that the CpfC of *S. aureus* and the SHP-1 of mammalian cells may be two independent target sites for SC-43, please see lines 527 to 531 of the revised manuscript.

What were the initial indications that CpfC is the cellular target? In the paper it should be mentioned the lead observations that led to CpfC identification (e.g., genetic resistance selection, metabolomics showing accumulation of coproporphyrin precursors, affinity capture/pull-down, phenotypic rescue by heme). If this was a chemical genetics screen or unbiased proteomics result, please summarize the key experiments and data that prioritized CpfC.

Response: Thank you very much for the reviewers' comments. We mainly rely on the antibacterial phenotypes and antibacterial spectra (Effective against Gram-positive bacteria used in this study but ineffective against Gram-negative bacteria, even when used in combination with outer membrane disruptors, please see Table 1 of the revised manuscript) to make judgments on whether

the candidate proteins with statistical significance from the TPP experiment are potential targets. Therefore, two conditions need to be met for a protein to be considered a potential target. Firstly, it must be a necessary protein for Gram-positive bacteria used in this study. Secondly, it is very likely to be non-essential for Gram-negative bacteria (we have ruled out the possibility that SC-43 is ineffective against Gram-negative bacteria due to the obstruction of the outer membrane by using outer membrane inhibitors). Therefore, the proteins that do not meet these two conditions were excluded from our study for mechanism studies, please see lines 154 to 159 of the revised manuscript.

We attempted to isolate the spontaneous drug-resistant strains of bacteria against SC-43, but failed. This indicates that bacteria are not prone to developing resistance to SC-43 on the one hand, and it also prevented us from obtaining genetic evidence on the other hand. Therefore, in the revised manuscript, we added the content about overexpression of *cpfC*. The results showed that after overexpressing of *cpfC*, the minimum inhibitory concentration (MIC) of the bacteria against SC-43 increased by 16-fold times, indicating the correlation between CpfC and the antibacterial activity of SC-43. Please see table 1 and lines 172 to 175 of the revised version.

To clarify the effects of SC-43 on the metabolic pathways, we conducted additional metabolomics experiments, which revealed that in addition to the porphyrin metabolism pathway closely related to CpfC, the O-antigen nucleotide sugar biosynthesis and lysine degradation pathways also be affected by SC-43. Therefore, we proposed that SC-43 has multiple effects on bacteria, please see lines 508 to 511 and Fig. 3A of the revised manuscript.

Moreover, metabolomics results indicate that SC-43 reduces the content of coproporphyrin precursors rather than causing accumulation (Fig. 3B). We speculate that this might be a compensatory measure taken by the bacteria due to the adverse effects of the accumulation of porphyrin metabolites on their survival. These have been explained in the discussion section, please see lines 483 to 486 of the revised manuscript.

For the murine wound model: indicate dosing schedule and route. Discuss the implication of heme in serum/tissue for efficacy in systemic vs. topical applications.

Response: Thank you very much for the reviewers' comments. We have added a schematic diagram showing the dosing schedule and route of administration, please see Fig. 6A of the revised manuscript.

We completely agree with the reviewer's suggestion regarding the possible influence of heme in serum and tissues on the antibacterial activity of SC-43. We have included a discussion of this part in the revised manuscript, please see lines 535 to 538 of the revised manuscript. We are extremely grateful for the reviewers' comments, which have significantly improved the quality of our manuscript.

Docking is useful, but co-crystallization of SC-43 with SaCpfC (or soaking) would be much stronger evidence of the binding mode. If a co-crystal was not obtained, state this limitation. At minimum, provide docking score, major interactions, and whether the predicted contacts agree with the mutant binding data.

Response: Thank you very much for the reviewers' comments. We completely agree with the comments. The failure to obtain the co-crystallization structure is a limitation of our work. Meanwhile, we performed molecular docking studies for SC-43 against the wild-type SA_{CpfC} protein and its mutants (Y12A, H181A, E263A) using the extra precision (XP) mode of Glide in the Schrödinger suite.

As shown in Appendix Fig. S2, the binding affinity, as reflected by the Glide GScore, was weakened for all mutants compared to the wild-type protein (GScore = -5.505). This computational finding is consistent with our experimental Surface Plasmon Resonance (SPR) data (Table 2 in the main text), which also showed reduced affinities for the mutants. The analysis of the binding poses reveals the structural basis for this decrease. In the Y12A mutant complex, the mutation to alanine abolishes both a critical hydrogen bond and a π - π stacking interaction that were observed with the tyrosine residue in the wild-type protein. Similarly, the binding poses for the H181A and E263A mutants show the loss of key interactions present in the wild-type complex. The disruption of these specific interactions provides a plausible explanation for the lower binding affinities measured by SPR. Please see Appendix Fig. S2 and lines 389 to 394 in the revised manuscript for the description of these results.

The hem biosynthesis has been very well studied in the group of Saraiva and they also identified hem-biosynthesis inhibitors. The papers should be discussed.

Lobo, S.A., Scott, A., Videira, M.A., Winpenny, D., Gardner, M., Palmer, M.J., Schroeder, S., Lawrence, A.D., Parkinson, T., Warren, M.J., and Saraiva, L.M. (2015) Staphylococcus aureus haem biosynthesis: characterisation of the enzymes involved in final steps of the pathway. Mol Microbiol 97: 472-487.

Mendes, S.S., Marques, J., Mesterhazy, E., Straetener, J., Arts, M., Pissarro, T., Reginold, J., Berscheid, A., Bornikoel, J., Kluj, R.M., Mayer, C., Oesterhelt, F., Friaes, S., Royo, B., Schneider, T., Brotz-Oesterhelt, H., Romao, C.C., and Saraiva, L.M. (2022) Synergetic Antimicrobial Activity and Mechanism of Clotrimazole-Linked CO-Releasing Molecules. ACS Bio Med Chem Au 2: 419-436.

Videira, M.A.M., Lobo, S.A.L., Silva, L.S.O., Palmer, D.J., Warren, M.J., Prieto, M., Coutinho, A., Sousa, F.L., Fernandes, F., and Saraiva, L.M. (2018) Staphylococcus aureus haem biosynthesis and acquisition pathways are linked through haem monooxygenase IsdG. Mol Microbiol 109: 385-400.

Videira, M.A.M., Lobo, S.A.L., Sousa, F.L., and Saraiva, L.M. (2020) Identification of the sirohaem biosynthesis pathway in Staphylococcus aureus. FEBS J 287: 1537-1553.

Response: We are extremely grateful for the reviewer's reminders. We carefully studied Saraiva's article and focused on discussing the work of this team in the heme synthesis pathway, particularly

in the hemoem synthesis process. Another aspect is about the interaction between IsdG and CpfC, which in turn affects heme synthesis. However, for the article published in ACS Bio Med Chem Au, we found that its mechanism was not related to the synthesis of heme. Therefore, we did not cite it in the revised manuscript. For the work of IsdG, in addition to the articles recommended by the reviewer, we also included the articles that were recently published (*J Inorg Biochem.* 2025;269:112878). This article conducted an in-depth study on the interaction between IsdG and CpfC.

20th Feb 2026

Dear Dr. Huang,

Thank you for the submission of your revised manuscript to EMBO Molecular Medicine. I am pleased to inform you that we will be able to accept your manuscript pending the following final amendments:

1) Authors:

- We note that you currently have together with you, a total of 4 co-corresponding authors. Is that correct? Do you confirm equal contribution of these 4 people, able to take full responsibility for the paper and its content? While there is no limit per se to the number of co-corresponding authors, 4 is rare, and may not reflect as intended to the community.

- Please provide institutional email addresses for the co-corresponding authors Wei Huang and Quanming Zou in our submission system and the title page of the manuscript.

2) Structural data:

- All publications must be accompanied by deposition of both the atomic coordinates and the structure-factor amplitudes in the appropriate database (PDB or NAKB). In the case of low-resolution structures for which only a chain trace is reported, a set of α positions and structure-factor amplitudes may be sufficient.

- Please provide the official wwPDB validation report.

3) Please consider revising the title to: SHP-1 agonist SC-43 limits Methicillin-resistant *Staphylococcus aureus* infection through inhibition of heme biosynthesis

4) In the main manuscript file, please do the following:

- Please address all comments suggested by our data editors listed below:

o Data availability statement:

1. Please note that the specific URL for PXD073199 dataset is not provided in the data availability statement.

2. Please note that reviewer access codes for PXD073199, PXD073397 datasets are provided in the manuscript.

o Figure legends:

1. Please note that figure titles for figures 1-5 is not provided. This needs to be rectified.

2. Please note that the exact p values are not provided in the legends of figures 1E, 2A-D; 3B, 6C, D.

3. Please indicate the statistical test used for data analysis in the legend of figure 3A.

4. Please note that the measure of center for the error bars needs to be defined in the legends of figures 1E, 2A-D; 3B, 6C, D.

- Remove all figures and move their legends to the end of the manuscript, after the references.

- Move Tables 1 and 2 to the end of the manuscript text, after the main figure legends.

- Indicate in legends exact n and exact p values, not a range, along with the statistical test used. To keep the figures "clear" some authors found providing an Appendix table Sx with all exact p-values preferable. You are welcome to do this if you want to.

- Please remove Reagents and Tools Table and uploaded it as a separate file. Correct the reference for primer sequences to Appendix Table S3 (see below for updated table numbering).

- In data availability statement please use the following format to report the accession number of your deposited data:

[data type]: [full name of the resource] [accession number/identifier] ([doi or URL or identifiers.org/DATABASE:ACCESSION])

Please check "Author Guidelines" for more information.

<https://www.embopress.org/page/journal/17574684/authorguide#availabilityofpublishedmaterial>

- Please correct the reference citation in the reference list. Where there are more than 10 authors on a paper, 10 will be listed, followed by "et al.". Also, please remove DOIs. Please check "Author Guidelines" for more information.

<https://www.embopress.org/page/journal/17574684/authorguide#referencesformat>

5) Funding: Please merge it with Acknowledgements.

6) Tables:

- Please rename Tables S2 and S3 to Dataset EV1 and Dataset EV2. Correct the nomenclature in the table legends, file names and citations in the manuscript text.

- Please upload "Figure Source Data Table 2" as Dataset EV3 with the table legend and cite it at the appropriate place in the main manuscript text.

7) Appendix: Please remove blank pages and add a table of contents with page numbers on the front page. Correct nomenclature to "Appendix Figure S1" etc. Add Tables S1, S4 and S5 to the appendix file as "Appendix Table S1-S3". Please update their callouts in the main text. The final version of the appendix file should be uploaded as a PDF.

8) Synopsis:

- Synopsis image: Please do not use smaller sized Figure 7 as a synopsis image. Provide a new, simplified image as a visual abstract to highlight the main finding of your study and upload it as a high-resolution jpeg file 550 px-wide x 300-600 pixels high to illustrate your article.

- Synopsis text: Please provide a short standfirst (maximum of 300 characters, including space) and redefine the bullet points to better summarise the paper. Upload the synopsis text as a .doc file. Please write the bullet points to summarise the key NEW findings. They should be designed to be complementary to the abstract - i.e. not repeat the same text. We encourage inclusion

of key acronyms and quantitative information (maximum of 30 words / bullet point). Please use the passive voice.

9) As part of the EMBO Publications transparent editorial process (see our Editorial at <http://embomolmed.embopress.org/content/2/9/329>), EMBO Molecular Medicine will publish online a Review Process File (RPF) to accompany accepted manuscripts. This file will be published in conjunction with your paper and will include the anonymous referee reports, your point-by-point response and all pertinent correspondence relating to the manuscript. Let us know if you want to remove or not any figures from it prior to publication. Please note that the Authors checklist will be published at the end of the RPF.

10) Please provide a point-by-point letter INCLUDING my comments as well as the reviewer's reports and your detailed responses (as Word file).

I look forward to reading a new revised version of your manuscript as soon as possible.

Yours sincerely,

Zeljko Durdevic

Zeljko Durdevic
Senior Editor
EMBO Molecular Medicine

*** Instructions to submit your revised manuscript ***

When preparing your revised manuscript, please refer to our guidelines: <https://link.springer.com/journal/44321/submission-guidelines#cms-Revised-submissions>. We perform an initial quality control of all revised manuscripts before re-review; failure to include requested items will delay the evaluation of your revision.

We require:

2) Individual production quality figure files as .eps, .tif, .jpg (one file per figure). For guidance, download the 'Figure Guide PDF': <https://media.springernature.com/original/springer-cms/rest/v1/content/27825798/data/v1>.

3) A .docx formatted letter INCLUDING the reviewers' reports and your detailed point-by-point responses to their comments. As part of the EMBO Press transparent editorial process, the point-by-point response is part of the Review Process File (RPF), which will be published alongside your paper.

4) A complete author checklist, which you can download from our author guidelines. Please insert information in the checklist that is also reflected in the manuscript. The completed author checklist will also be part of the RPF.

5) Please note that all corresponding authors are required to supply an ORCID ID for their name upon submission of a revised

manuscript.

6) It is mandatory to include a 'Data Availability' section after the Materials and Methods. Before submitting your revision, primary datasets produced in this study need to be deposited in an appropriate public database, and the accession numbers and database listed under 'Data Availability'. Please remember to provide a reviewer password if the datasets are not yet public.

7) For data quantification: please specify the name of the statistical test used to generate error bars and P values, the number (n) of independent experiments (specify technical or biological replicates) underlying each data point and the test used to calculate p-values in each figure legend. The figure legends should contain a basic description of n, P and the test applied. Graphs must include a description of the bars and the error bars (s.d., s.e.m.).

9) Our journal encourages inclusion of *data citations in the reference list* to directly cite datasets that were re-used and obtained from public databases. Data citations in the article text are distinct from normal bibliographical citations and should directly link to the database records from which the data can be accessed. In the main text, data citations are formatted as follows: "Data ref: Smith et al, 2001" or "Data ref: NCBI Sequence Read Archive PRJNA342805, 2017". In the Reference list, data citations must be labeled with "[DATASET]". A data reference must provide the database name, accession number/identifiers and a resolvable link to the landing page from which the data can be accessed at the end of the reference.

12) Author contributions: You will be asked to provide CRediT (Contributor Role Taxonomy) terms in the submission system. These replace a narrative author contribution section in the manuscript.

13) A Conflict of Interest statement should be provided in the main text.

14) Every published paper includes a 'Synopsis' to further enhance discoverability. Synopses are displayed on the journal webpage and are freely accessible to all readers. They include a short stand first (maximum of 300 characters, including space) as well as 2-5 one-sentences bullet points that summarizes the paper. Please write the bullet points to summarize the key NEW findings. They should be designed to be complementary to the abstract - i.e. not repeat the same text. We encourage inclusion of key acronyms and quantitative information (maximum of 30 words / bullet point). Please use the passive voice. Please attach these in a separate file or send them by email, we will incorporate them accordingly.

15) Include a Reagents and Tools Table as part of the Methods section, which can be downloaded from our author guidelines.

Photos 400-800 DPI

*Additional important information regarding figures and illustrations can be found at
<https://media.springernature.com/original/springer-cms/rest/v1/content/27825798/data/v1>

***** Reviewer's comments *****

Referee #1 (Comments on Novelty/Model System for Author):

The authors have adequately addressed this reviewer's comments.

Referee #2 (Remarks for Author):

The authors have addressed my prior comments.

The authors addressed the remaining editorial issues.

16th Mar 2026

Dear Dr. Huang,

We are pleased to inform you that your manuscript is accepted for publication and is now being sent to our publisher to be included in the next available issue of EMBO Molecular Medicine.

You may qualify for financial assistance for your publication charges - either via a Springer Nature fully open access agreement or an EMBO initiative. Check your eligibility: <https://link.springer.com/journal/44321/how-to-publish-with-us>

Zeljko Durdevic
Senior Editor
EMBO Molecular Medicine

>>> Please note that it is EMBO Molecular Medicine policy for the transcript of the editorial process (containing referee reports and your response letter) to be published as an online supplement to each paper. If you do NOT want this, you will need to inform the Editorial Office via email immediately. More information is available here: <https://link.springer.com/partners/embo-press/editorial-policies#Peer%20review>